# Global warming due to loss of large ice masses and Arctic summer sea ice

Nico Wunderling [1,2,3✉], Matteo Willeit[1], Jonathan F. Donges [1,4] & Ricarda Winkelmann [1,2✉]

Several large-scale cryosphere elements such as the Arctic summer sea ice, the mountain glaciers, the Greenland and West Antarctic Ice Sheet have changed substantially during the last century due to anthropogenic global warming. However, the impacts of their possible future disintegration on global mean temperature (GMT) and climate feedbacks have not yet been comprehensively evaluated. Here, we quantify this response using an Earth system model of intermediate complexity. Overall, we find a median additional global warming of 0.43 °C (interquartile range: 0.39−0.46 °C) at a $CO_2$ concentration of 400 ppm. Most of this response (55%) is caused by albedo changes, but lapse rate together with water vapour (30%) and cloud feedbacks (15%) also contribute significantly. While a decay of the ice sheets would occur on centennial to millennial time scales, the Arctic might become ice-free during summer within the 21st century. Our findings imply an additional increase of the GMT on intermediate to long time scales.

[1] Earth System Analysis, Potsdam Institute for Climate Impact Research (PIK), Member of the Leibniz Association, Potsdam D-14473, Germany. [2] Institute of Physics and Astronomy, University of Potsdam, Potsdam D-14476, Germany. [3] Department of Physics, Humboldt University of Berlin, Berlin D-12489, Germany. [4] Stockholm Resilience Centre, Stockholm University, Stockholm, SE 10691, Sweden. ✉email: nico.wunderling@pik-potsdam.de; ricarda.winkelmann@pik-potsdam.de

Extensive changes have been observed in large-scale cryo-sphere elements during the last decades such as the Arctic summer sea ice, mountain glaciers, the Greenland and West Antarctic Ice Sheet[1–5].

From the late 1970s to the mid-2000s, the Arctic summer sea ice area has declined by more than 10% per decade, as satellite measurements reveal[1]. If this trend continues, the Arctic could become ice-free in summer for the first time within the 21st century. Projections with CMIP-5[6] (Coupled Model Inter-comparison Project Phase 5) models show that this could be the case as early as 2030 to 2050 for higher emission scenarios such as RCP8.5 (Representative Concentration Pathway)[7]. Some GCMs (global circulation models) show an ice-free Arctic for the first time within this century also for the moderate emission scenarios at a warming of 1.7 °C above pre-industrial[8,9]. Furthermore, observations reveal that the Arctic summer sea ice declines faster than expected in experiments from GCMs[1].

At the same time, mountain-glaciers world-wide have retreated, with an average weight equivalent ice loss of approxi-mately 250 ± 30 Gt per year between 1901 and 2009[2,10]. This translates, in the same time span, into a loss of 21% of the gla-ciated volume of mountain glaciers worldwide, excluding (Sub-) Antarctic peripheral glaciers, as found in model simulations[11]. During this time, it is estimated that approximately 600 glaciers have disappeared and many more are likely to follow in the future (IPCC-AR5, Chapter 4[6]). 36 ± 8% of today's glacier mass is already committed to be lost in response to past greenhouse gas emissions[12] and it has been found that many mountain glaciers are currently in disequilibrium and will be subject to further ice loss[13].

Moreover, both the West Antarctic and the Greenland Ice Sheet have lost mass at an accelerating pace in the past decades[3–5]. With progressing global warming, ice loss from the polar ice sheets and subsequent sea-level rise is expected to further increase[14,15]. Beyond a critical temperature threshold, large parts of the Greenland Ice Sheet might melt, accelerated by positive feedbacks such as the ice-albedo and melt-elevation feedbacks[16,17]. From model simulations, this threshold temperature is suggested to range between 0.8 and 3.2 °C above pre-industrial levels[18].

Parts of the West Antarctic Ice Sheet might already have crossed a point of instability: the grounding lines of several gla-ciers in the Amundsen basin are rapidly retreating and have likely become unstable, causing sustained ice discharge from the entire basin which could lead to more than 1 m of global sea-level rise[19]. Similar dynamics might be induced in other parts of the Antarctic Ice Sheet and could eventually lead to its complete disintegration under unmitigated climate change[20].

Anthropogenic climate change has already caused a rise in global mean temperature (GMT) by 0.9 °C comparing 1850–1900 to 2006–2015[21], with observable impacts on the cryosphere ele-ments mentioned above[6]. It has also been suggested that these regions are likely to change dramatically with ongoing climate warming and some of these changes are suspected to possess some degree of irreversibility[22,23].

Following these recent developments of the cryosphere com-ponents, it seems possible that they might be lost at lower tem-peratures than commonly thought, potentially as low as 1.5 °C above pre-industrial levels[23]. The disintegration of these elements is associated with feedbacks that impact back on GMT, for instance via a change in albedo, clouds or lapse rate, among others, which has not been quantified comprehensively so far. Therefore, we assess the additional global warming caused by disintegration of the Greenland Ice Sheet, the West Antarctic Ice Sheet, the mountain glaciers and the Arctic summer sea ice. Although the Arctic summer sea ice is implemented in more complex Earth system models and its loss part of their simulation results (e.g. in CMIP-5), it is one of the fastest changing cryo-sphere elements whose additional contribution to global warming is important to be considered. Therefore, we compute and sepa-rate its contribution to GMT increase. On the other side, the temperature feedbacks of ice sheets like Greenland, West Ant-arctica and mountain glaciers are not yet fully integrated in assessments such as CMIP-5.

We base our simulations on the Earth system model of inter-mediate complexity, CLIMBER-2[24,25] because it is computa-tionally efficient and allows a systematic analysis of the decay of the cryosphere components. CLIMBER-2 includes atmosphere, ocean, sea ice, vegetation and land-ice model components and has been applied extensively to understand past and future climate changes[26,27].

In large ensembles of equilibrium model simulations, con-strained by fast climate feedbacks strength from global circulation models[28] (see "Methods"), we compare the long-term GMT change in idealised scenarios, where the cryosphere elements are removed, to scenarios where they remain intact. The uncertainty in the additional warming in our simulations is constrained by the uncertainty of the feedback strength in the GCM simulations which we used to mimic the more complex behaviour of GCMs[28] (Supplementary Fig. 1). To change the feedback strengths, we alter CLIMBER-2 model parameters that act on the strength of the feedbacks themselves, particularly in the structure of the troposphere and the clouds (atmospheric changes) as well as in the snow albedo (see Supplementary Table 1). With reasonably altered parameters in CLIMBER-2, we arrive at an equilibrium climate sensitivity of 2.0–3.75 °C for our ensemble leading to smaller temperature responses than the full range from CMIP-5 (2.0–4.7 °C) or CMIP-6 (1.8–5.6 °C) would[29]. Details on the calibration process are given in the methods section: uncertainty estimates.

In our experiments the state of the Greenland Ice Sheet, the West Antarctic Ice Sheet and mountain glaciers is simply pre-scribed in the model and affects both, ice cover and topography. In our simulations for the Arctic summer sea ice, the albedo during the summer months (June, July, August) is lowered to average values for open ocean waters instantaneously similar to Blackport and Kushner[30], while keeping the computation of ice-covered areas dynamic, such that the experiment does not violate energy and water conservation.

In this study, we find that global warming is amplified by the decay of the Earth's cryosphere as expected from theory and quantify the contribution of each of the four cryosphere compo-nents. We further separate the GMT response into contribu-tions from albedo, lapse rate, water vapour and clouds in terms of perturbation of the net radiation at the top of the atmosphere[31]. Here, we focus on the purely radiative effects and neglect fresh-water contributions to feedbacks and warming. Thus, our esti-mates are long-term equilibrium responses when the large ice masses are disintegrated. However, transient warming responses would be reduced due to freshwater input from the West Ant-arctic and Greenland Ice Sheet on centennial time-scales[32–35].

## Results

**Additional global and regional warming.** We consider several different climate scenarios, with atmospheric $CO_2$ concentrations ranging from the pre-industrial 280 ppm up to 700 ppm and run the model forward until it reaches equilibrium. If not stated otherwise, our findings are shown for a reference simulation at a fixed $CO_2$ concentration of 400 ppm in equilibrium after 10,000 years. 400 ppm corresponds to an equilibrium GMT increase of 1.5 °C above pre-industrial in CLIMBER-2 simulations. Upon this, we evaluate the additional regional and global warming

caused by the large-scale loss of the Arctic sea ice during summer, mountain glaciers, and the polar ice sheets. While this ad-hoc loss of the ice masses poses a hypothetical scenario, it allows us to separate the additional warming through the ice-climate feedbacks from other effects. In our experiments, we report the median value of the ensemble and the brackets represent the interquartile range unless stated otherwise.

In our simulations, we find that global warming is increased by the decay of the Earth's cryosphere. The disintegration of the Arctic summer sea ice and the retreat of mountain glaciers, the Greenland and the West Antarctic Ice Sheets together cause an additional GMT increase of 0.43 °C (0.39–0.46 °C) for a baseline-scenario of 1.5 °C warming above pre-industrial levels, which translates into an additional warming of 29% (26–31%).

Locally, the loss of each element induces a very strong warming signal, which is consistent with previous studies on polar and Arctic amplification[36,37]. Local warming around the cryosphere components is up to 5 °C stronger, particularly around Greenland and West Antarctica (Fig. 1a). However, the ice loss causes significant warming also in lower latitudes, with values of 0.2 °C around the equator.

The warming results from our simulations are consistent in magnitude and polar amplification with past warm periods, particularly the Mid-Pliocene Warm Period, during which the large ice sheets were at least partially disintegrated[38,39]. Still, the distribution among the feedback processes in these paleoclimate states remains uncertain.

Under ongoing global warming, further ice loss is to be expected for all of the four cryosphere components considered here; however, the corresponding time scales differ by several orders of magnitude. While substantial ice loss from Greenland or Antarctica might be triggered by anthropogenic climate change within the current century, these changes would manifest over several centuries to millennia[15]. Ice-free Arctic summers on the other side might already occur in the next decades[1,7,9]. Therefore, we also consider the regional warming caused solely by the loss of the Arctic summer sea ice (Fig. 1b). The additional warming in the Arctic region on a yearly average accounts for more than 1.5 ° C regionally and for 0.19 °C globally. The meltdown of the Arctic sea ice and its regional warming effect is also simulated by CMIP-5 runs dependent on the future anthropogenic $CO_2$ forcing scenarios, the RCP scenarios[6,9].

With CLIMBER-2, we are able to distinguish between the respective cryosphere elements and can compute the additional warming resulting from each of these (Fig. 2). The additional warmings are 0.19 °C (0.16–0.21 °C) for the Arctic summer sea ice, 0.13 °C (0.12–0.14 °C) for GIS, 0.08 °C (0.07–0.09 °C) for mountain glaciers and 0.05 °C (0.04–0.06 °C) for WAIS, where the values in brackets indicate the interquartile range and the main value represents the median. If all four elements would disintegrate, the additional warming is the sum of all four individual warmings resulting in 0.43 °C (0.39–0.46 °C) (thick dark red line in the Fig. 2). Our results regarding the amount of warming are of comparable magnitude to previous efforts computed for late Pliocene realisations (PRISM) of the ice sheets[40,41]. Both studies show a pronounced warming in the proximity of the locations where ice is removed, which is in good agreement with our results (see Fig. 1 and Supplementary Fig. 2).

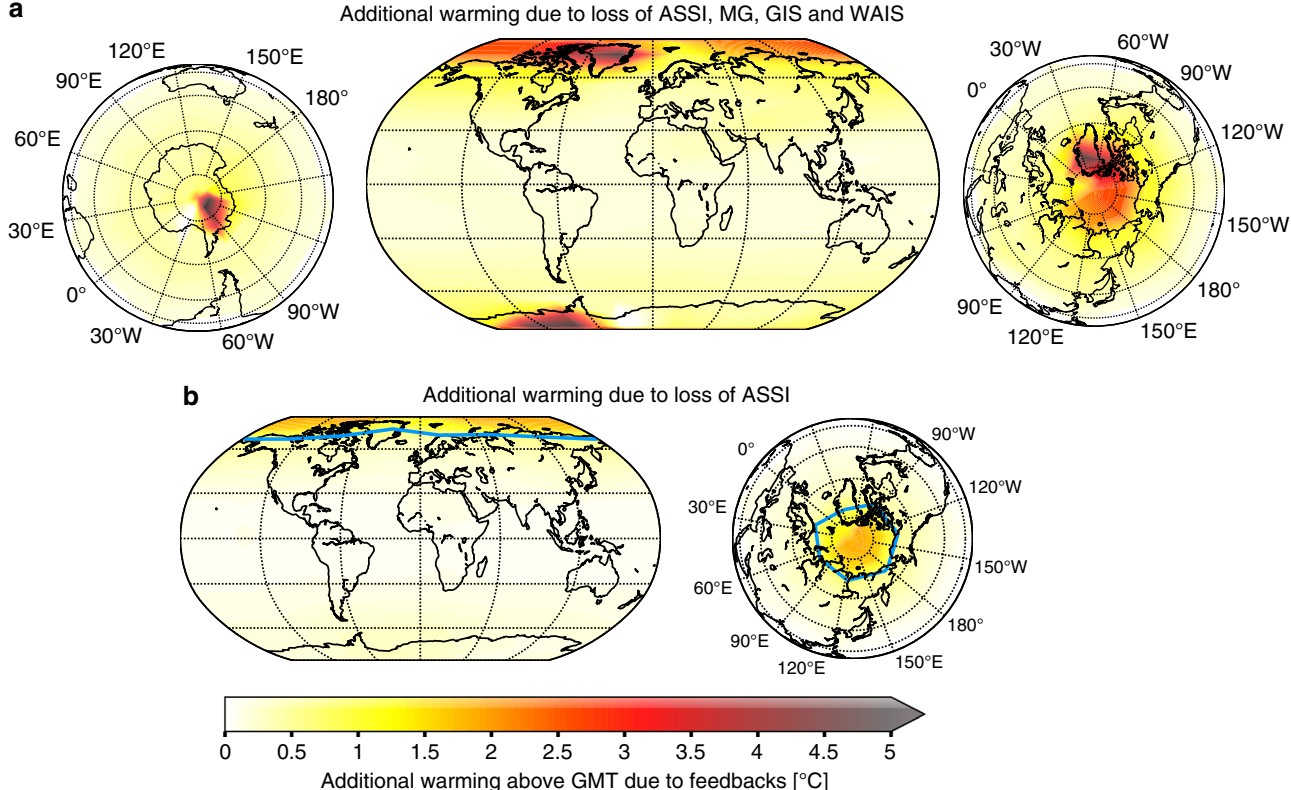

**Fig. 1 Regional warming due to feedbacks. a** Regional warming for the whole Earth if Arctic summer sea ice (ASSI) in June, July and August, mountain glaciers (MG), Greenland Ice Sheet (GIS) and West Antarctic Ice Sheet (WAIS) vanish at a global mean temperature of 1.5 °C above pre-industrial. **b** Same as in (**a**) with an additional zoom-in of the Arctic region if only the Arctic summer sea ice vanishes, which might happen until the end of the century. The light blue line indicates the region of removed Arctic summer sea ice extent, where its concentration in CLIMBER-2 is 15% or higher. In all panels, the average additional warming on top of 1.5 °C is shown in absolute degree.

The disintegration of all elements at the same time can very closely be approximated by the sum of single elements disintegrated indicating that their effects on GMT add up linearly. This can be found in Fig. 3, where we also show the warming for $CO_2$ concentrations from 280 to 700 ppm. Fig. 2 highlights the additional warming of 1.5 °C above pre-industrial.

**Warming from the Arctic summer sea ice**. We obtain that the warming results are independent from the $CO_2$ concentration

### Additional global warming commitment due to ice loss

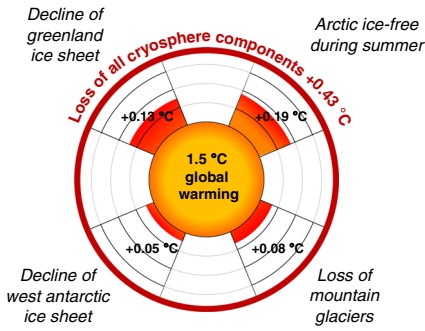

**Fig. 2 GMT increase through disappearance of cryosphere elements.** The additional warming for the cryosphere components is shown for a scenario consistent with global warming levels of 1.5 °C. Radially outward, the temperature anomaly is displayed which arises from the disappearance of the cryosphere elements. The thick dark red line indicates the maximum effect of additional warming in case all cryosphere elements lose stability. All values are the medians of the ensemble.

forcing between 280 and 700 ppm apart from the Arctic summer sea ice (see Fig. 3a), which shows a decreasing additional warming for higher $CO_2$ concentrations (Fig 4). This can, in turn, be explained: In CLIMBER-2 simulations we find, with increasing prescribed $CO_2$ concentrations corresponding to increasing GMT, that the Arctic summer sea ice area declines in a linear way, which was also found in observational records[42] and in GCM simulations[9]. For a $CO_2$ concentration of 400 ppm corresponding to 1.5 °C in CLIMBER-2 above pre-industrial GMT levels, the additional warming is 0.19 °C (0.16–0.21 °C). The actual minimal sea ice cover observed by NERSC (Nansen Environmental & Remote Sensing Center) as an average area from 1979 to 2006 is on the order of 5.5–6.5 × 10$^6$ km$^2$ which would correspond to a warming of approximately 0.15 °C in our simulations (see Fig. 4). In Supplementary Fig. 3, we show the sea ice area over the course of 1 year for the control and the perturbed run.

**Radiative perturbations at the top of the atmosphere**. For each cryosphere element, we are able to deconvolve the net change of radiative perturbations at the top of the atmosphere into several components that affect the radiative balance of the Earth: water vapour, clouds, lapse rate and albedo. These factors can be quantified in CLIMBER-2 (Table 1).

The values for water vapour, lapse rate and clouds in Table 1 can to a very good approximation directly be interpreted as feedback factors once they are divided by the respective warming, e.g., by 0.43 °C in case all investigated cryosphere elements are removed. However, it is important to note that the perturbation arising from albedo changes is both, a forcing and a feedback. The forcing component originates from the prescribed removal of the cryosphere elements. On the other side, the feedback component

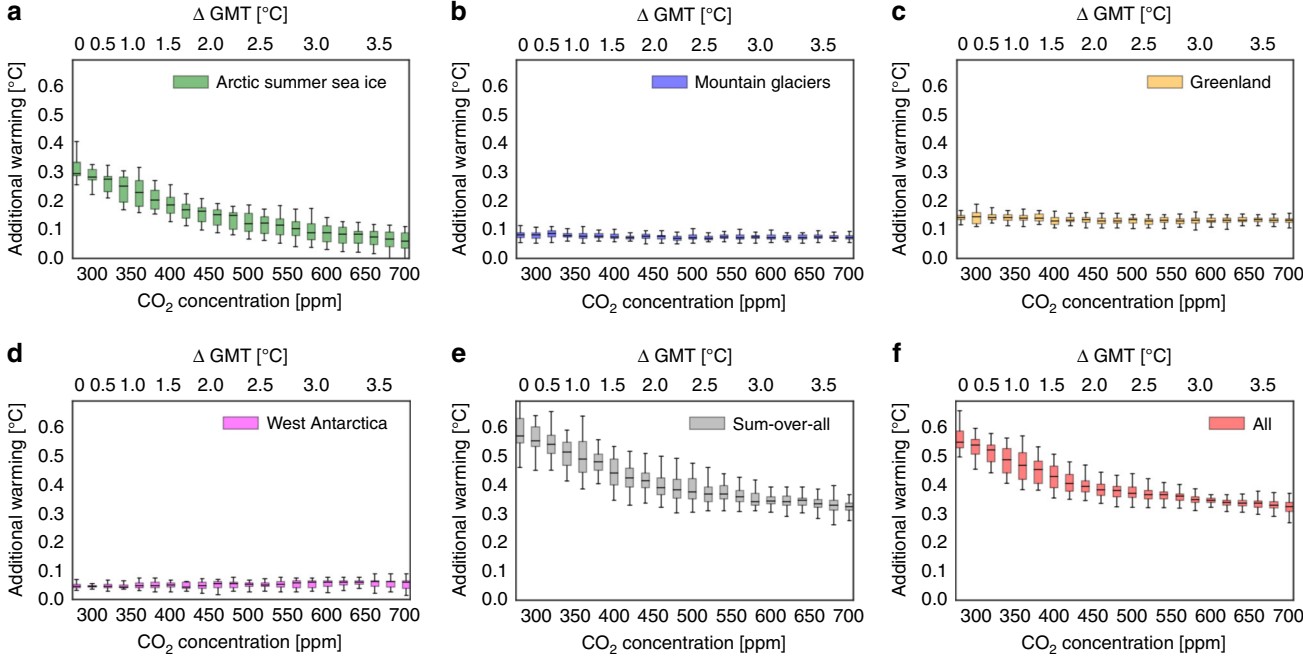

**Fig. 3 Linearity of additional warming due to disintegration of cryosphere elements.** Additional warming plotted against $CO_2$ concentration. Disintegration of of cryosphere components separately for (**a**) the Arctic summer sea ice, (**b**) the mountain glaciers, (**c**) the Greenland Ice Sheet, (**d**) the West Antarctic Ice Sheet, (**e**) the sum of all additional warmings from the separately disintegrated cryosphere elements and (**f**) the disintegration of all four elements at the same time. The grey bars match the red bars within their errors which means, according to CLIMBER-2, that the warming effect of singular disintegrated cryosphere elements can linearly be added up to the effect of all four elements disintegrated at the same time. Here we show median, interquartile range and full ensemble spread for each $CO_2$ concentration. The upper horizontal axis shows the temperature increase above pre-industrial, where a least-square fit converting $CO_2$ concentration to temperature with python's function scipy.optimize.curve_fit was used. The respective fitted temperatures arise from full ensemble simulations at prescribed $CO_2$ concentrations, but without removed cryosphere elements.

derives from responses of the surface albedo to the additional warming as for instance through changes in the extent of snow covered area or changes in vegetation cover. Thus both, the feedback and the forcing contribute to the measured radiative perturbation quantified in Table 1.

Change in surface albedo is the dominant additional radiative perturbation for each considered cryosphere element. It is mainly caused by the albedo change of large ice-covered areas from ice to other non ice-covered surface types, but also by other land cover changes. In total around 55% of the radiative perturbations can be attributed to the change of the albedo.

Two more additional radiative perturbations which are evaluated together as they are anti-correlated are the lapse rate and the water vapour fast climate feedback[28,31]. The lapse rate change arises from non-uniform temperature changes in the vertical atmospheric column and subsequent changes in outgoing longwave radiation. The water vapour change describes the capacity of the air to sustain water vapour in the air. The capacity

to sustain water vapour is increased by 7% per degree of warming as can be computed using the Clausius–Clapeyron equation. Since the GMT is increasing through the removal of the cryosphere elements, the air can sustain more water vapour which then in turn leads to an additional warming. Together, the additional radiative perturbation of water vapour and lapse rate combine for approximately 30% of the complete radiative perturbation.

For the cloud feedbacks, the IPCC AR5 and newer studies hypothesised that the feedback from clouds is likely positive[6,43] as we also find here. It is responsible for 15% of the total radiative perturbation.

Within our experimental setting, it can be expected that the radiative perturbation from albedo changes is very high due to the prescribed removal of the respective cryosphere element. However, the radiative perturbation related to different fast climate feedbacks such as water vapour, lapse rate and clouds also play an important role as drivers of additional warming. Together they account for more than 40% of the total radiative perturbation on average.

Similar investigations on the additional radiative perturbation from albedo changes have been performed for the removal of Arctic sea ice. For a removal of one month during summer an additional radiative perturbation of 0.3 W/m$^2$ is reported[44] which is in good agreement with Flanner et al. (2011)[45]. We find a slightly higher value of 0.49 W/m$^2$ for albedo plus clouds value when the Arctic summer sea ice is removed (Table 1). This value probably is higher since we have low sea ice for approximately five months (Supplementary Fig. 3) in our perturbed experiments instead of one as in Hudson[44], but parts of the deviation might also be due to the slightly different experimental setup.

In Supplementary Fig. 4a, we show the latitudinal distribution of the additional radiative perturbation at the top of the atmosphere. The contribution from albedo as well as from lapse rate and water vapour are higher in polar regions and thus contribute to polar amplification which is also apparent in the corresponding zonal mean surface warming (see Supplementary Fig. 4b). On the other hand, the additional cloud feedback does not strongly contribute to polar amplification in our simulations. These trends for clouds and albedo have also been found by other studies[36,46]. Further studies mention that the lapse rate feedback plays a major role in polar amplification[47]. This seems to be the case here as well (see Supplementary Fig. 4a), but we can only make this statement for the combined feedbacks of lapse rate and water vapour since we do not separate them in our analysis.

## Discussion

Our results concern short and long term effects on GMT due to the disintegration of cryosphere elements which experienced significant changes within the last decades and are likely to also change strongly in the future due to global warming.

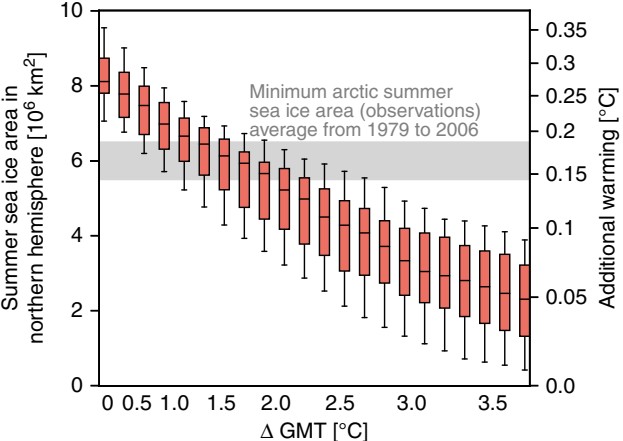

**Fig. 4 Additional warming due to meltdown of Arctic summer sea ice.** Box whiskers plot of global mean temperature (ΔGMT) versus Arctic summer sea ice area with error boxes (error bars) representing the interquartile range (full spread) of the ensemble at the according GMT over the CLIMBER-2 ensemble runs. The additional warming when the Arctic summer sea ice disappears is represented by a second y-axis computed via a least-square fit from the corresponding summer sea ice area. The relationship between summer sea ice area and additional warming is slightly nonlinear. This means that a doubling of the ice area does not quite translate into a doubling of the additional warming. The x-axis shows ΔGMT above pre-industrial computed via a GMT-CO$_2$ concentration least-square fit. The shaded area shows the mean Arctic sea ice area as observed by NERSC (Nansen Environmental & Remote Sensing Center) from 1979 to 2006, where the uncertainty indicates one standard deviation: 6.0 ± 0.5 × 10$^6$ km$^2$.

**Table 1 Drivers of warming as seen from the top of the atmosphere.**

| Cryosphere element | LR + WV [W/m$^2$] | Clouds [W/m$^2$] | Albedo [W/m$^2$] | All changes [W/m$^2$] |
|---|---|---|---|---|
| ASSI | 0.20 (0.17–0.23) | 0.08 (0.07–0.09) | 0.41 (0.35–0.47) | 0.69 (0.59–0.79) |
| GIS | 0.14 (0.13–0.16) | 0.06 (0.05–0.07) | 0.22 (0.20–0.25) | 0.43 (0.39–0.47) |
| WAIS | 0.05 (0.04–0.05) | 0.04 (0.03–0.05) | 0.10 (0.08–0.11) | 0.18 (0.16–0.21) |
| MG | 0.09 (0.08–0.10) | 0.04 (0.03–0.05) | 0.16 (0.14–0.17) | 0.28 (0.26–0.32) |
| All | 0.45 (0.41–0.49) | 0.17 (0.16–0.19) | 0.72 (0.66–0.78) | 1.35 (1.22–1.46) |

*ASSI* Arctic summer sea ice, *GIS* Greenland Ice Sheet, *WAIS* West Antarctic Ice Sheet, *MG* Mountain glaciers.
The additional radiative perturbation for the fast climate feedbacks as evaluated in CLIMBER-2 at a global warming of 1.5 °C above pre-industrial for disintegration of the respective element given as changes in W/m$^2$. The values are given as median and interquartile range (in brackets) of the ensemble. The "LR + WV" column represents the lapse rate and water vapour additional radiative perturbation column together as they are anti-correlated and thus not independent[57]. Note that the albedo forcing values refer to both, a forcing and a feedback. The forcing part is the removal of the cryosphere components and the feedback part comprises changes in vegetation and snow cover in response to the additional warming.

On shorter time scales, the decay of the Arctic summer sea ice would exert an additional warming of 0.19 °C (0.16–0.21 °C) at a uniform background warming of 1.5 °C (=400 ppm) above pre-industrial. On longer time scales, which can typically not be considered in CMIP projections, the loss of Greenland and West Antarctica, mountain glaciers and the Arctic summer sea ice together can cause additional GMT warming of 0.43 °C (0.39–0.46 °C). This effect is robust for a whole range of $CO_2$ emission scenarios up to 700 pm and corresponds to 29% extra warming relative to a 1.5 °C scenario.

In fact, some feedbacks will also be at play before the complete disintegration of the large ice sheets, for instance due to increased ice-drainage from the Amundsen region in West Antarctica[19,48,49]. Furthermore, it has been shown for WAIS and GIS that transgressing their critical thresholds is likely not reversible due to hysteresis effects[18,50,51].

The additional commitment to global warming that we study here represents a long-term, mean-field effect which is separated from possible direct interactions between the elements such as the freshwater input into the thermohaline circulation from the large ice sheets. In other words, the disintegration of the ice sheets has a direct increasing temperature impact on the GMT via the feedbacks quantified here.

## Methods

**Earth system model**. For our analysis, we use the Earth system model of inter-mediate complexity (EMIC) CLIMBER-2[24,25] on a coarse spatial resolution of 10 × 52° (lat × lon) resolution. CLIMBER-2 includes a 2.5-D dynamical-statistical atmosphere and a multi-basin, zonally averaged ocean model including sea ice as well as a dynamic model of the terrestrial biosphere. CLIMBER-2 also includes a model for ice sheets, a global carbon cycle model and an atmosphere surface interaction coupler, which are not used in this study since ice sheets and atmospheric $CO_2$ are prescribed in our experiments. In CLIMBER-2, changes in the cloud fraction are possible. Apart from that, cloud top height can change following changes in the height of the tropopause. The cloud optical thickness parameterisation includes a dependence on the cumulus cloud fraction in addition to a prescribed increase of optical thickness with latitude. With this representation of clouds, CLIMBER-2 is able to reproduce the planetary albedo as observed from CERES (see Supplementary Fig. 5)[52]. We benefit from the use of an EMIC as it is highly computationally efficient and allows for a systematic analysis of the impact of disintegration of the cryosphere elements on GMT. With CLIMBER-2 we are able to distinguish different feedbacks and are able to run a robustness analysis using systematic parameter studies. CLIMBER-2 is a good representative of other EMICs[53].

**Model initialisation**. In preparation of the model runs, we set up the ice sheets inbuilt in CLIMBER-2. For distinguishing the West and East Antarctic Ice Sheet, we created a mask based on the Antarctic drainage basins[54]. We also included a mountain glacier mask with data from the Randolph glacier inventory[55]. Since we are interested in the climatological behaviour of the disintegration of one or more of the cryosphere elements, we artificially change the setup of CLIMBER-2 depending on which element we remove: In case of WAIS and GIS, the topography of the ice sheet itself is removed together with the ice sheet as the height of the ice sheet is several thousand metres thick and thus might play an important role on the feedbacks. The albedo is replaced by the albedo of bare land or ocean (where appropriate) at first, but can then change freely into any kind of vegetation or snow cover during the simulation run. For our simulations, isostatic rebound is neglected.

For the Arctic summer sea ice and the mountain glaciers, the topography is not taken into account as either the height of the ice or the spacial extent of high thickness regions is very low. To remove the Arctic summer sea ice during the summer months (June, July and August: JJA), the surface covered by sea ice is darkened and the albedo in this region is replaced by the ocean albedo. With this procedure the energy conservation law is not violated since the ice is not just removed and still retains its function as boundary layer between ocean and atmosphere. Thus we are able to compute the effect of summer sea ice in an energetically self-consistent manner. Note that CLIMBER-2 is mass conserving. Our procedure is similar to the experimental setup of Blackport and Kushner[30], who also reduce albedo values of the sea ice instantaneously. They do this for the whole year and all sea ice compared to our setup, where the albedo is changed only in the northern hemisphere in the summer months.

**Model calibration**. To emulate the behaviour of more complex general circulation models (GCMs) we created a model ensemble by perturbing several parameters with the target to cover the range of strength of the fast climate feedbacks found by

Soden and Held[28] using an ensemble of GCMs. Equally, this could have been done with the feedbacks stated in the IPCC assessment report 5 (AR5), but changes in the reported feedback strengths are small except for the cloud feedback which is less well constrained in AR5 (see IPCC on page 819 for a direct comparison between AR5 values and the values given in Soden and Held[28]). Thus, our ensemble and our results can be expected to stay the same. The fast climate feedbacks include the water vapour, the lapse rate, the cloud and the albedo feedback. Each of our 39 ensemble members, that we end up with, is constructed from a pair of simulations: one control run at 280 ppm and one perturbed run at a $CO_2$ doubling of 560 ppm. We then compute the magnitude of the fast climate feedbacks between these pairs of runs (see Supplementary Fig. 1a). Here, we evaluate the feedbacks using the partial radiation perturbation method[31,56]. In this method partial derivatives of model top of the atmosphere radiation with respect to changes in model parameters (such as water vapour, lapse rate and clouds) are determined by diagnostically rerunning of the model radiation code.

The water vapour feedback added to the lapse rate feedback is supposed to lie in the range of 0.8–1.2 W/m²/K. These two feedbacks are evaluated together as they are correlated negatively[28,57]. The cloud feedback is supposed to range between 0.3 and 1.1 W/m²/K and the albedo feedback between 0.2 and 0.45 W/m²/K. Furthermore, we put a constraint on the minimal summer sea ice cover in the northern hemisphere to 1.5–6.5 km² (see Supplementary Fig. 1d). In Soden and Held[28], the albedo value is constraint to values between 0.2 and 0.4 W/m²/K, but in our calibration run, it is necessary to increase the upper limit to 0.45 W/m²/K since vegetation shifts are considered and otherwise the ensemble gets distorted to small summer sea ice values in the control run.

On top of the fast climate feedbacks, we require each ensemble member (each pair of runs) to possess an equilibrium climate sensitivity above 1.5 and below 4.5 °C, where the equilibrium climate sensitivity is the global warming per doubling of atmospheric $CO_2$ concentration (see Supplementary Fig. 1b). It is important to note that our ensemble members span the range from 2.0 to 3.75 °C. This leads to smaller temperature response ranges than the full range from 1.5 to 4.5 °C would. Furthermore, a last constraint is applied at a $CO_2$ concentration of 280 ppm. The temperature difference between the runs with perturbed parameters and the reference run with unperturbed parameters (brackets in Supplementary Table 1) should be less or equal than ±1.0 °C (see Supplementary. Fig. 1c). After the application of all these constraints, we find 39 pairs of runs that match our restrictions.

For covering the uncertainty ranges of the feedbacks we perturb parameters (within their experimental uncertainty range) influencing lapse rate together with the water vapour, cloud and albedo feedbacks similarly to Deimling et al.[57] (Supplementary Table 1). With this procedure, we are able to reconstruct the uncertainty ranges of the four fast climate feedbacks stated in Soden and Held[28] fairly well.

**Uncertainty estimates**. We used these 39 calibrated runs, which also represent the uncertainty of our results, as initialisation for our large-scale ensemble simulations. For each of the cryosphere elements, i.e., WAIS, GIS, Arctic summer sea ice and mountain glaciers, as well as all together, we performed the following experiments: (i) Control runs: the respective cryosphere element(s) is/are kept and (ii) experiment runs: removed cryosphere element(s).

We performed the experiments in (i) and (ii) for different atmospheric $CO_2$ concentrations as external forcing. We chose the $CO_2$ concentration parameter since it is the one which is most probably increasing in future climate change scenarios. Each of the experiments is performed as a long term equilibrium run for 10,000 simulation years with today's boundary conditions, i.e., astrophysical parameters like eccentricity and obliquity, and fixed $CO_2$ concentration. The results are taken as the mean over the last 4000 simulated years since this cancels out minor fluctuations in the equilibrium state. In the end we subtract the experimental run (ii) from the control run (i) to retrieve the temperature difference. Since we are reporting these differences between perturbed (experimental) and control run throughout the main manuscript, the uncertainties given as interquartile ranges are small, also compared to the calibration (see Supplementary Fig. 1). This means that our CLIMBER-2 ensemble is robust against the same perturbations in the cryosphere components. We constructed our ensemble aiming at covering a range of sensitivities and different strengths of the feedbacks by the variation of the parameters in Supplementary Table 1.

## Data availability

The data that support the findings of this study are available from the corresponding author upon reasonable request.

## Code availability

There is no comprehensively documented code for the Earth system model CLIMBER-2 available owing to a lack of comprehensive technical description, but the code is available upon request from M.W.

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

## Acknowledgements

This work has been carried out within the framework of the IRTG 1740/TRP 2015/50122-0 funded by DFG and FAPESP. N.W. and R.W. acknowledge their support. M.W. acknowledges support from the BMBF through the project PalMod. J.F.D. is grateful for financial support by the Stordalen Foundation via the Planetary Boundary Research Network (PB.net), the Earth League's EarthDoc programme, and the European Research Council Advanced Grant project ERA (Earth Resilience in the Anthropocene; grant ERC-

2016-ADG-743080). We are thankful for support by the Leibniz Association (project DominoES). The authors gratefully acknowledge the European Regional Development Fund (ERDF), the German Federal Ministry of Education and Research and the Land Brandenburg for supporting this project by providing resources on the high performance computer system at the Potsdam Institute for Climate Impact Research.

## Author contributions

N.W., M.W., J.F.D. and R.W. designed the study and wrote the text including revisions. N.W. conducted the model simulation runs and prepared the figures. M.W. designed the model calibration and prepared CLIMBER-2 for simulations.

## Funding

## Competing interests

The authors declare no competing interests.
