## [Peer Review File · Nature Communications]

Editorial Note: The figure on page 18 in this Peer Review File is reproduced with permission from John Wiley & Sons, Inc. The figure on page 19 in this Peer Review File is reproduced with permission from Elsevier.

Reviewers' comments:

Reviewer #1 (Remarks to the Author):

Dear Editor, dear Authors,

I finalised my reading of the paper by Wunderling et al. "Additional global warming commitment due to crossing critical thresholds within the Earth's cryosphere", submitted to Nature communications.

It is an a priori suitable manuscript, with a good question, a robust method to diagnose the answer, and an answer clear enough for the targetted journal.

It is likely an original synthesis going beyond previous knowledge.

Yet it might be a bit early to publish this manuscript as is.

This is because clarification on several aspects is required. The manuscript is not detailed enough for understanding the context, what has been done and evaluating the robustness of what is claimed.

Best wishes,

Clarification is needed:

1) On the evaluation of feedbacks...

- How do you evaluate feedbacks (there are several means to do so, see Bony et al, 2006) ?

- Why would you ignore Planck feedback ? This might be obvious, but all classical studies mention it (Bony et al. 2006 for example). In addition, recent studies on the attribution of polar amplification to climate feedbacks (Pithan & Mauritsen, 2014; Goosse et al., 2018; Stuecker et al., 2018) mention this as a key process. Hence I think it would be worth to explain why it is missing from your analysis.

2) On the state of the art...

What do we know of cryosphere-climate feedbacks ? Where does it come from ? What is your contribution wrt state of the art ?

Key contributions from the following references are in my view missing:

- the IPCC AR4 8th chapter by Randall et al, with a specific section of cryospheric feedbacks

- the paper by Blackport and Kushner in J. Clim (401-417, 2016). These guys actually use a very close experimental setup to yours for sea ice.

- the paper of Stuecker et al., Nature Climate Change 2018 on polar amplification.

- the review on polar feedbacks by Goosse et al in Nature Communications 2018.

3) On the experimental setup...

- What is the horizontal resolution of the model ?
- What do you do with freshwater once the cryospheric elements are removed ?
- How different your sea ice setup is different from Blackport and Kushner (2016) ?
- What type of sea ice loss do you achieve ? I cannot tell with certitude whether you loose all sea ice or some sea ice ? Could you show plots depicting how is sea ice lost in supplementary material ?

4) On the results

- To which extent does your evaluation of cryospheric feedback departs from that of the IPCC report for snow and ice feedback factor of 0.26 ± 0.08 W/m²/K of AR4 ?
- To which extent the answer that the albedo feedback dominates the response is self-contained in your perturbation technique ? This could be particularly true for the case of sea ice ?
- To which extent your conclusion that the ice-albedo feedback dominates is expected ?
- How would you frame your results in the recent attribution of polar amplification to temperature feedbacks by Pithan and Mauritsen (2014) and Stuecker et al (2018). Planck feedback and LR feedbacks have been proposed as the largest contributors ? Is the cryosphere extra contribution to global warming related to the polar amplification ? Can you tell ?

--- A few extra specific comments

- L. 257-258 don't understand what is meant ?
- L. 150-152 What does this sentence bring here ?
- Fig. 3 Would it make sense to add observations and CMIP5 ensemble here

Reviewer #2 (Remarks to the Author):

Summary of claims:

The paper finds that the disappearance of the 4 currently most affected cryosphere elements (WAIS=West-Antartic Ice Sheet, GIS=Greenland Ice Sheet, ASSI=Arctic Summer Sea Ice, MG=Mountain Glacier) would contribute an additional increase to global mean temperature (GMT) of about 1/3 C. The claim is based on equilibrium analysis of the intermediate-complexity climate model CLIMBER-2. The reference GMT increase for the 1/3 C number is 1.5..2.0 C (the Paris Climate target). The study was conducted very systematically for CO₂ concentrations from 280 to 700ppm (FigS2). The spatial distribution of the additional warming is sharply localized (up to 2.5 C in the West Antarctic and Greenland). The analysis also includes the effect of removing only one cryoelement at a time (for each of the 4).

Novelty/interest in wider field:

Since the climate targets and the notion of climate sensitivity use the equilibrium as their reference, the paper may contribute to the public and scientific debate on climate change. The paper raises the distinct possibility of an additional 1/3 C warming being built into the already very

ambitious target scenario, if the most vulnerable cryo-elements disappear. The results are certainly original. They are specific to CLIMBER-2, but this model is a good representative of other EMICS (old comparison study from 2005: Petoukhov, V., Claussen, M., Berger, A. et al. *Climate Dynamics* (2005) 25:363. doi:10.1007/s00382-005-0042-3). The prior calibration should also make the results less model specific.

A remarkable feature of the results (that is not commented on) is that the uncertainty in the results is very low. For example, the headline result is stated as 0.37+/-0.03 C. This is remarkable since the model was calibrated to run with an ensemble spread of feedback parameters to cover the entire (much larger) range of sensitivities and internal feedbacks reported by Soden & Held. (If I understood correctly.)

The headline result on GMT and spatial distribution of temperature is very much in the expected range. A criticism would be that the scenario comparison between complete removal vs undisturbed presence of (each of) the 4 cryoelements may not be the center of the debate. The more intricate research of what causes (for example) discrepancies between observed ASI trends and simulations is not addressed (and not aimed for) by this study.

Robustness of evidence:

The study is very comprehensive in terms of varied parameters and scenarios (an advantage of restricting to an EMIC).

One issue not requiring new studies but far better explanation is the table 1 with the feedback factors. I am clear how Soden & Held extracted their feedback factor estimates from models (referenced in Fig S1), and I assume that the same methodology was used for the measurements in the calibration runs. (Presumably with the 4 cryoelements prescribed as present?) However, it is unclear to me how the numbers in table 1 are arrived at or what they even mean. The quantities have the same names as those in Fig S1, but (some) are much larger with smaller uncertainty. Does, for example, the 2.43+/-0.21 W/m²/K in the [ASSI,Albedo] row/column mean that a simulation without ASSI shows an albedo feedback factor of 2.43 (thus much larger than the 0.2 in a world with ASSI)? Why would a simulation without sea ice have a so much stronger albedo feedback than one with sea ice (given that there would be no ice that could vary in response to Delta T to change the albedo)?

These feedback numbers are puzzling but they may give a more important stimulus to the internal scientific discussion than the headline result.

Also, the calibration is poorly explained. The methods section mentions two calibration runs but Fig S1 shows many more data points. The methods section also mentions variation of feedback parameters. So, I assume that there was a sequence of 280,560ppm CO₂ run pairs with varying parameters?

Minor presentation issues are that Fig 1 has the steepest gradients (and strongest effects) in the most distorted parts of the map. Showing two pole projections may be more suitable (or at least an area-preserving projection).

Figure S2 is much more informative and straightforward to understand than Figure 2 because it gives the wider context, shows the low uncertainty and the conversion between CO₂ and Delta T (I assume that this conversion is for the control runs with cryoelements present).

Figure 3 should show the additional warming as a sequence of error bars (on the right axis) instead of the hard-to-judge color coding.

Potential for influence:

The headline figure will add to the debate, possibly even at the public level. (However, there the fact that the scenario was complete removal of the cryoelement will probably get lost.) The feedback factors, if properly explained and if they mean what I think they mean, could have significant influence in the scientific community.

Appropriateness and validity of (statistical) analysis, reproducibility:

These were computer experiments. The error bars are small and the trends are very clear and agree with established theory such that I do not see a major problem with statistics. The calibration of the model parameters to cover a range of feedback factors and sensitivities should also make the result repeatable with other climate models of similar computational complexity.

Reviewer #3 (Remarks to the Author):

Review of "Additional global warming commitment due to crossing critical thresholds within the Earth's cryosphere" submitted to Nature Communications by Nico Wunderling, Matteo Willeit, Jonathan F. Donges, and Ricarda Winkelmann.

This manuscript aims to quantify the additional global warming commitment due to the loss of different components of the cryosphere (Arctic sea ice, mountain glaciers, and the ice sheets of Greenland and West Antarctica). Each of these components is removed within a model of intermediate complexity (an EMIC with a simplified atmosphere with parameterized feedbacks and a zonal average ocean) and the additional warming, at specified CO₂ levels, is reported in long simulations run to equilibrium (10,000 years). Mountain glaciers and ice sheets are removed from the model by changing elevation and albedo while sea ice loss is initiated by changing the sea ice albedo. The authors report that at 400 ppm, corresponding to an equilibrium warming of about 1.5K above pre-industrial in the model, the loss of these cryosphere components would add 0.37K \pm 0.03K. Their interpretation is that crossing cryosphere tipping points would lead to a further rise of global temperature that could push us beyond the Paris 1.5K to 2K warming target even with strong emissions reductions.

Recommendation: The manuscript is well written and the results are easy to follow. However, I see several major issues with the work that, in my opinion, make it unsuitable for publication in Nature Communications. One major issue involves the overall premise of the research – there is a fundamental confusion about what is and is not a cryosphere tipping point and, moreover, I do not believe the results have any relevance for Paris temperature targets as claimed. Other major issues involve technical aspects of whether results are realistic. Below I provide details about this assessment should the authors choose to continue this line of work.

Major comments

M1: Several specific cryosphere components are chosen based on the perception that they are "at risk of transgressing a critical threshold which could cause large-scale, possibly irreversible changes". The authors then explore the global temperature response to the removal of each cryosphere component individually and together and claim that this additional warming could have bearing on the Paris targets. I believe this premise to be flawed in several ways.

Firstly, not all of the cryosphere components chosen are at risk of passing a critical threshold or tipping point beyond which loss would be irreversible. Summer Arctic sea ice loss in particular is thought to be reversible with respect to global temperature (e.g., doi: 10.1175/JCLI-D-14-00654.1, 10.5194/tcd-5-2349-2011, 10.1029/2011GL048739, doi:10.1029/2010GL045698, 10.1038/nature09653). Moreover, the contribution of sea ice loss to global temperature is already

included in the Earth System Model simulations used to quantify the relationship between cumulative carbon and global temperature and thus is already included in the Paris targets. By artificially removing sea ice the authors seem to be implicitly making the argument that GCMs/ESMs are missing a mechanism by which sea ice is lost irreversibly at low warming levels (e.g., 1.5K), but they do not describe or justify such a mechanism thus making the choice seem quite arbitrary. One could just as easily induce the loss of all low clouds or all snow within a model at 1.5K and quantify the additional warming that would ensue, but such experiments would not be meaningful or have any bearing on temperature targets.

The loss of mountain glaciers is also, to my knowledge, a reversible process. This experiment seems better justified by the fact that much of the glacier loss will happen this century yet may not be included in many GCMs/ESMs and thus is a potential source of additional warming that has relevance for temperature targets. If this is the first study to quantify the global surface albedo feedback and additional warming associated with mountain glaciers then this should be noted. I suspect that it is not, in which case results should be compared to previous efforts.

The complete loss of the Greenland and West Antarctic Ice Sheets may be irreversible, but the timescales for ice sheet loss are centuries to millennia meaning that there is little relevance for the Paris temperature targets. It has been known for some time that the loss of ice sheets would induce additional warming, with estimates of the Earth System Sensitivity (ESS) ranging from about 30% to 100% above the Equilibrium Climate Sensitivity (ECS) which excludes slow feedback changes (e.g., doi: 10.1038/ngeo706, 10.2174/1874282300802010217, 10.1038/nature11574). I was struck by how small the additional warming from GIS and WAIS were (0.14K and 0.06K, respectively, on top of the 1.5K baseline), and found the manuscript lacking in its discussion of these results in the context of the many previous studies.

In general, the 1.5K to 2K Paris temperature target has been chosen to avoid the loss of the GIS and WAIS. If the authors then want to claim that their results have bearing on these temperature targets then they have to provide some evidence that that the irreversible loss of these cryosphere components will occur at temperatures lower than 1.5K to 2K, but I see no such discussion. Overall this study has the feel of a modelling exercise with little to no connection to reality.

M2: The accuracy of the results (additional warming commitment) depends critically on the feedbacks as represented in CLIMBER-2. I recognize that the authors have put much time and effort into spanning a range of cloud and lapse rate feedback parameters to represent a range of ECS values (1.5-4.5 K). But I see some technical issues that need to be addressed or explained.

- Key to the results is the strength of the feedbacks induced by the cryosphere changes. The magnitude of the LR+WV and cloud feedbacks (Table 1) seem reasonable compared to GCMs, theory and observations. However, the albedo feedback seems far too large with values ranging from 1.7 W/m²/K (loss of GIS) to 2.4 W/m²/K (loss of summer Arctic sea ice). To put this in perspective, the surface albedo feedback in GCMs under abrupt CO₂ quadrupling, where most summer Arctic sea ice is lost, is typically about 0.4 W/m²/K. Table 1 also shows the sum of all feedbacks to generally exceed 3.2 W/m²/K, which is the value of the Planck feedback and thus the maximum allowed feedback value to produce a stable climate. Clearly either something has gone wrong in these feedback calculations or CLIMBER-2 has a very strange climate. The authors should examine the surface albedo feedback in the baseline simulations (without induced cryosphere loss) to see if it is at all realistic and reexamine their calculations.

- The very small range of uncertainty for the global warming response to loss of Earth's cryosphere (0.37 +/- 0.03 K) is simply not believable. How can this small range be reconciled with the factor of three range of ECS spanned by the model (1.5-4.5 K), for instance?

Minor comments

- L29-31: I encourage the authors to define what they mean if they are to use the term 'tipping element'. Are the changes of these cryosphere components reversible with respect to global temperature or not, and how do we know? This seems key to motivating this line of work (see above).

- L173-174: I do not understand this sentence. There are several factors contributing to positive cloud feedback in GCMs, not just associated with cloud amount and not just in mid to high latitudes (see for example doi: 10.1038/nclimate3402)

- L185-187: As noted above, Arctic sea ice loss is already included in ESMs used to quantify the maximally allowed CO2 emissions.

- L225-227: Is the ice sheet albedo replaced with the lbedo of ocean, bare land, or land with vegetation? Is isostatic rebound accounted for in the response to loss of GIS or WAIS?

- L276-279: This behavior in GCMs usually has to do with deep convective variability in the region of the Weddell sea polynya. Is this what you mean by 'the same behavior' in your model. I doubt it. I suspect it is a model artefact associated with your idealized zonal average ocean.

Response to the reviewers' comments

Dear Editor, Dear Reviewers,

Thank you very much for your insightful comments and the chance to improve our manuscript substantially. We are very happy that our work is of potential interest and found your comments and suggestions very helpful in revising the manuscript.

In light of your reviews and to better cope with the issues raised, we decided to re-run the whole ensemble of climate model simulations with a new calibration. With this new calibration, we ran in total 1716 new ensemble members over a CO₂-concentration range from 280 to 700 ppm. In the new ensemble of simulations, we now consider a larger range of Arctic summer sea-ice area. The overall results are consistent with our original ensemble and the conclusions remain robust.

Furthermore, we substantially revised the manuscript - major changes include the following:

1. An additional constraint is now imposed on the minimum Arctic sea-ice area which lies between 1.5 - 6.5 mio. km² in our new ensemble. Furthermore, we increased the range for the albedo feedback value to 0.2 - 0.45 W/m²/K and put a constraint on the reference temperature to +1°C as opposed to +2°C in the original analysis (see supplementary Fig. S1 and methods section of the main manuscript).
2. We invested major efforts in the re-design of our figures as well as in the requested more thorough investigation of the fast climate feedbacks (see main text in the section about climate feedbacks referred to Table 1 as well as supp. Fig. S1 & S3).
3. We added a discussion of the results on regional warming due to disintegration of cryosphere elements and included a figure to the supplementary material (Fig. S4), where we extended our simulation to the removal of the whole Antarctic Ice Sheet to be able to compare the regional and global warming with previous efforts.

Besides these major revisions, we answer the specific issues you raised in a point by point fashion in the following paragraphs (please see below) and marked our changes in the revised version of our manuscript.

Sincerely yours,

Nico Wunderling, Matteo Willeit, Jonathan F. Donges and Ricarda Winkelmann

Reviewer #1 (Remarks to the Author):

Dear Editor, dear Authors,

I finalised my reading of the paper by Wunderling et al. "Additional global warming commitment due to crossing critical thresholds within the Earth's cryosphere", submitted to Nature communications.

It is an a priori suitable manuscript, with a good question, a robust method to diagnose the answer, and an answer clear enough for the targeted journal.

It is likely an original synthesis going beyond previous knowledge.

Yet it might be a bit early to publish this manuscript as is.

This is because clarification on several aspects is required. The manuscript is not detailed enough for understanding the context, what has been done and evaluating the robustness of what is claimed.

Best wishes,

Clarification is needed:

1) On the evaluation of feedbacks...

- How do you evaluate feedbacks (there are several means to do so, see Bony et al, 2006) ?

Feedbacks are evaluated using the partial radiation perturbation method (Wetherald and Manabe, 1988, Bony et al., 2006). In this method partial derivatives of model TOA radiation with respect to changes in model parameters (such as water vapor, lapse rate, and clouds) are determined by diagnostically rerunning of the model radiation code. (see main manuscript II. 291 ff.)

- Why would you ignore Planck feedback ? This might be obvious, but all classical studies mention it (Bony et al. 2006 for example). In addition, recent studies on the attribution of polar amplification to climate feedbacks (Pithan & Mauritsen, 2014; Goosse et al., 2018; Stuecker et al., 2018) mention this as a key process. Hence I think it would be worth to explain why it is missing from your analysis.

We are of course not ignoring the Planck feedback, which is fully active in our model. The Planck feedback is obviously of fundamental importance, but it is quite well known and there is almost no spread between models in its estimated strength. The Planck feedback depends on latitude and that is why it has an impact on polar amplification, but we do not think that adding a discussion about that would provide additional value to our paper. (see II. 109 ff.)

2) On the state of the art...

What do we know of cryosphere-climate feedbacks ? Where does it come from ? What is your contribution wrt state of the art ?

Key contributions from the following references are in my view missing:

Thanks for pointing out these additional important sources on feedback processes. We have read and included the following sources and computed the zonal mean temperature and the latitudinal distribution of the feedback processes (in supplementary Fig. S3) such that we are able to compare the results with the papers that you recommended to comment on (IPCC AR4, Stuecker et al., 2018, Goosse et al., 2018 and Pithan & Mauritsen, 2014). (see II. 214 ff.)

- the IPCC AR4 8th chapter by Randall et al, with a specific section of cryospheric feedbacks
- the paper by Blackport and Kushner in J. Clim (401-417, 2016). These guys actually use a very close experimental setup to yours for sea ice.
- the paper of Stuecker et al., Nature Climate Change 2018 on polar amplification.
- the review on polar feedbacks by Goosse et al in Nature Communications 2018.

3) On the experimental setup...

- What is the horizontal resolution of the model ?

The resolution of the model is 10x52° (lat x lon) and respectively 18x7 cells, see Methods section Earth system model (see I. 248)

- What do you do with freshwater once the cryospheric elements are removed ?

Here, we are computing the maximum commitment to global mean temperature in case all of the ice sheets melted down. Although we performed experiments with freshwater forcing in the beginning, we were here interested in the equilibrium response of the cryosphere elements to global mean temperature. This is why we impose no freshwater forcing from the melting ice sheets, but only remove them. In principle we acknowledge that the new equilibrium state could depend on the model response to a transient

freshwater forcing. However, during the 10,000 years of each ensemble member within our simulations, CLIMBER-2 is mass conserving. (see II. 111 ff.)

- How different your sea ice setup is different from Blackport and Kushner (2016) ?

Our procedure is similar to the experimental setup of Blackport and Kushner, who also reduce albedo values of the sea ice instantaneously. However, they change the albedo of the sea ice for the whole year in the northern and southern hemisphere, while we only change albedo in the northern hemisphere in the summer months. But with this way, Blackport and Kushner (and we) are able to compute the impact of Arctic summer sea ice in an energetically self-consistent way.

We integrated this reference in our revised paper. Thanks for pointing us there. (see II 105 f. and II. 276 ff.)

- What type of sea ice loss do you achieve ? I cannot tell with certitude whether you lose all sea ice or some sea ice ? Could you show plots depicting how is sea ice lost in supplementary material ?

Yes, supplementary Figure S2 now shows the sea ice area in the northern hemisphere for the control and the perturbed (loss of all cryosphere elements) run over the course of one year at a CO₂ concentration of 280 ppm. In the perturbed run, we darken the albedo of the summer sea ice area during June, July and August. What we find is that in the summer sea ice is completely lost during August, September and October. On the other hand, the difference between control and perturbed run during spring months (March, April, May) is small. (see II. 179 f. and supp. Fig. S2)

4) On the results

- To which extent does your evaluation of cryospheric feedback departs from that of the IPCC report for snow and ice feedback factor of $0.26 \pm 0.08 \text{ W/m}^2/\text{K}$ of AR4 ?

Our ensemble covers a range of albedo feedback factors between 0.25 and 0.45 W/m²/K (Fig. S1a), which includes the mean IPCC estimate. However, most of the ensemble members have a slightly stronger albedo feedback, partly because our albedo feedback factor also includes changes in surface albedo resulting from vegetation shifts. (see II. 209-303 and supp. Fig. S1a)

- To which extent the answer that the albedo feedback dominates the response is self-contained in your perturbation technique ? This could be particularly true for the case of sea ice ? To which extent your conclusion that the ice-albedo feedback dominates is expected ?

It can be expected that the albedo feedback is the strongest feedback since the albedo of large areas on the Earth surface change from ice covered to other types of coverage. This is especially the case in polar regions, where most of the removed ice masses reside, see also supplementary Fig. S3. Thus, this is to some degree self-contained in our perturbation technique, especially in the case of the Arctic summer sea-ice. However, on top of the confirmation of this suspicion, we compute an actual quantification of how

important each of the four evaluated feedbacks is and how much of the warming can be expected to be attributable to these. We would see this quantification as one of the added values of this study. (see II. 190 ff. and supp. Fig. S3)

- How would you frame your results in the recent attribution of polar amplification to temperature feedbacks by Pithan and Mauritsen (2014) and Stuecker et al (2018). Planck feedback and LR feedbacks have been proposed as the largest contributors ? Is the cryosphere extra contribution to global warming related to the polar amplification ? Can you tell ?

Removing ice sheets or Arctic sea ice will of course lead to a warming in high latitudes, where the ice is removed. This will act to amplify the warming at the poles relative to the tropics and therefore contribute to polar amplification. But as we show in the paper, in this case the warming mainly results from a decreased surface albedo (see also supp. Fig. S3).

--- A few extra specific comments

- L. 257-258 don't understand what is meant ?

Rephrased - Hopefully it is now better understandable (see II. 304 ff.)

- L. 150-152 What does this sentence bring here ? Fig. 3 Would it make sense to add observations and CMIP5 ensemble here?

We integrated the observations from NSIDC (minimum sea-ice area from the average over 2001-2010) in Fig. 4 now and referred to them. (see Fig. 3)

Reviewer #2 (Remarks to the Author):

Summary of claims:

The paper finds that the disappearance of the 4 currently most affected cryosphere elements (WAIS=West-Antartic Ice Sheet, GIS=Greenland Ice Sheet, ASSI=Arctic Summer Sea Ice, MG=Mountain Glacier) would contribute an additional increase to global mean temperature (GMT) of about 1/3 C. The claim is based on equilibrium analysis of the intermediate-complexity climate model CLIMBER-2. The reference GMT increase for the 1/3 C number is 1.5..2.0 C (the Paris Climate target). The study was conducted very systematically for CO₂ concentrations from 280 to 700ppm (FigS2). The spatial distribution of the additional warming is sharply localized (up to 2.5 C in the West Antarctic and Greenland). The analysis also includes the effect of removing only one cryoelement at a time (for each of the 4).

Novelty/interest in wider field:

Since the climate targets and the notion of climate sensitivity use the equilibrium as their reference, the paper may contribute to the public and scientific debate on climate change. The paper raises the distinct possibility of an additional 1/3 C warming being built into the already very ambitious target scenario, if the most vulnerable cryo-elements disappear. The results are certainly original. They are specific to CLIMBER-2, but this model is a good representative of other EMICS (old comparison study from 2005: Petoukhov, V., Claussen, M., Berger, A. et al. *Climate Dynamics* (2005) 25:363. doi:10.1007/s00382-005-0042-3). The prior calibration should also make the results less model specific.

Thank you for hinting at this publication. It is now integrated in our manuscript. (see I. 257)

A remarkable feature of the results (that is not commented on) is that the uncertainty in the results is very low. For example, the headline result is stated as 0.37+/-0.03 C. This is remarkable since the model was calibrated to run with an ensemble spread of feedback parameters to cover the entire (much larger) range of sensitivities and internal feedbacks reported by Soden & Held. (If I understood correctly.)

This is true and how it was meant. We also see that this point needs better explanation and representation. This is why we now do not show standard deviations anymore in our temperature plots (and in the main text), but we refer to the whole ensemble now. This is why we give our errors as ranges for the whole ensemble now instead of the “+/-” notation. We use this procedure now as we think that no ensemble member should be treated special since all of them passed the calibration process.

For example the difference from the minimal value to the maximal value is larger (which is hidden by a simple standard deviation), for example at a CO₂ concentration of 400 ppm, the smallest ensemble member gives a warming of 0.36°C, the largest 0.51°C, for the removal of all four cryosphere elements (mean 0.42°C).

All in all, we think that the headline result of now 0.42°C (0.36 - 0.51°C) is reasonable (beforehand 0.37±0.03°C).

(see all results and especially II. 155 - 162)

The headline result on GMT and spatial distribution of temperature is very much in the expected range. A criticism would be that the scenario comparison between complete removal vs undisturbed presence of (each of) the 4 cryoelements may not be the center of the debate. The more intricate research of what causes (for example) discrepancies between observed ASI trends and simulations is not addressed (and not aimed for) by this study.

The focus of this study is the quantification of the commitment of additional global warming if the cryosphere elements vanish which are under danger within global warming levels of 2°C. The implications of this would be that climate change above a certain threshold of warming might be self-amplifying without further combustion of fossil fuels.

It is true that it would be intriguing for future studies to find an explanation for the reasons why Earth system model simulations of the Arctic summer sea ice reveal discrepancies to observations. However, we think that with our simulations, we cannot add up to this discussion, also because we are not using a full fledged Earth system model, but an EMIC.

Robustness of evidence:

The study is very comprehensive in terms of varied parameters and scenarios (an advantage of restricting to an EMIC).

One issue not requiring new studies but far better explanation is the table 1 with the feedback factors. I am clear how Soden & Held extracted their feedback factor estimates from models (referenced in Fig S1), and I assume that the same methodology was used for the measurements in the calibration runs. (Presumably with the 4 cryoelements prescribed as present?) However, it is unclear to me how the numbers in table 1 are arrived at or what they even mean. The quantities have the same names as those in Fig S1, but (some) are much larger with smaller uncertainty. Does, for example, the 2.43±0.21 W/m²/K in the [ASSI,Albedo] row/column mean that a simulation without ASSI shows an albedo feedback factor of 2.43 (thus

much larger than the 0.2 in a world with ASSI)? Why would a simulation without sea ice have a so much stronger albedo feedback than one with sea ice (given that there would be no ice that could vary in response to Delta T to change the albedo)?

These feedback numbers are puzzling but they may give a more important stimulus to the internal scientific discussion than the headline result.

For the calibration runs we performed a standard feedback analysis for CO2 doubling experiments using the partial radiation perturbation method to compute the feedback factors (e.g. Wetherald and Manabe, 1988). These feedback factors can be directly compared to results of CMIP models and we use this as one of our constraints for the model ensemble (Fig. S1a). For the calibration all 4 cryoelements are prescribed as present. (see II. 291 ff.)

The numbers in Table 1 were instead derived from a perturbed run with cryosphere elements removed (each separately and all together), meaning that we computed the feedback factors from the radiation perturbation at TOA when the cryosphere elements were removed.

We agree with the reviewer that the numbers presented in Table 1 could be misleading, because what is usually reported are the numbers of the feedback factors for a CO2 doubling. To avoid confusion, in Table 1 we now report the radiative forcing introduced by the removal of the cryosphere components through changes in albedo, water vapor, lapse rate and clouds. (see II. 185 ff. and II. 214 ff.)

In the revised Methods section we also describe in more detail how the feedback analysis is performed.

Also, the calibration is poorly explained. The methods section mentions two calibration runs but Fig S1 shows many more data points. The methods section also mentions variation of feedback parameters. So, I assume that there was a sequence of 280,560ppm CO2 run pairs with varying parameters?

Yes, this is how it was meant. We rephrased the methods section on model calibration and hope that is clearer and understandable now. (see methods section)

Minor presentation issues are that Fig 1 has the steepest gradients (and strongest effects) in the most distorted parts of the map. Showing two pole projections may be more suitable (or at least an area-preserving projection).

We added two pole projections to Figure 1 such that these regions are better visible and less distorted.

Figure S2 is much more informative and straightforward to understand than Figure 2 because it gives the wider context, shows the low uncertainty and the conversion between CO2 and Delta T (I assume that this conversion is for the control runs with cryoelements present).

Yes we agree. Figure S2 contains more information than our former Figure 2 which only includes the information for a specific CO2 concentration equal to 1.5°C of global warming. This is why we included Fig S2 in our main manuscript (Fig. 3 now).

Figure 3 should show the additional warming as a sequence of error bars (on the right axis) instead of the hard-to-judge color coding.

We now introduced a second y-axis instead of the colour bar in Fig. 3.

Potential for influence:

The headline figure will add to the debate, possibly even at the public level. (However, there the fact that the scenario was complete removal of the cryoelement will probably get lost.) The feedback factors, if properly explained and if they mean what I think they mean, could have significant influence in the scientific community.

Appropriateness and validity of (statistical) analysis, reproducibility:

These were computer experiments. The error bars are small and the trends are very clear and agree with established theory such that I do not see a major problem with statistics. The calibration of the model parameters to cover a range of feedback factors and sensitivities should also make the result repeatable with other climate models of similar computational complexity.

Reviewer #3 (Remarks to the Author):

Review of “Additional global warming commitment due to crossing critical thresholds within the Earth’s cryosphere” submitted to Nature Communications by Nico Wunderling, Matteo Willeit, Jonathan F. Donges, and Ricarda Winkelmann.

This manuscript aims to quantify the additional global warming commitment due to the loss of different components of the cryosphere (Arctic sea ice, mountain glaciers, and the ice sheets of Greenland and West Antarctica). Each of these components is removed within a model of intermediate complexity (an EMIC with a simplified atmosphere with parameterize feedbacks and a zonal average ocean) and the additional warming, at specified CO₂ levels, is reported in long simulations run to equilibrium (10,000 years). Mountain glaciers and ice sheets are removed from the model by changing elevation and albedo while sea ice loss is initiated by changing the sea ice albedo. The authors report that at 400 ppm, corresponding to an equilibrium warming of about 1.5K warming above pre-industrial in the model, the loss of these cryosphere components would add 0.37K \pm 0.03K. Their interpretation is that crossing cryosphere tipping points would lead to a further rise of global temperature that could push us beyond the Paris 1.5K to 2K warming target even with strong emissions reductions.

Recommendation: The manuscript is well written and the results are easy to follow. However, I see several major issues with the work that, in my opinion, make it unsuitable for publication in Nature Communications. One major issue involves the overall premise of the research – there is a fundamental confusion about what is and is not a cryosphere tipping point and, moreover, I do not believe the results have any relevance for Paris temperature targets as claimed. Other major issues involve technical aspects of whether results are realistic. Below I provide details about this assessment should the authors choose to continue this line of work.

We know that there is a discussion in the science community of how a tipping element should be defined and also feel that this should be clarified in our manuscript. This is why, we clarified in the manuscript that we use the definition of Lenton et al. (2008) in this study which does, for example, not require that a tipping element has hysteresis. This means that glaciers and the Arctic summer sea ice are counted as tipping elements (see also Schellnhuber et al., 2016). (see II. 38 ff. and II. 82 ff.)

Furthermore, we acknowledge that we need to be careful when framing our results in terms of their significance towards the Paris Agreement. We here want to make two statements. The first one is that the sea ice is relevant until 2100 because according to the observations it vanishes faster than experiments with global circulation models show (see also plot below adapted from Stroeve et al., 2012). The second important point is that, even if we achieve the Paris range until 2100 and limit the combustion of fossil fuels

after 2100 strictly, we might end up with a warmer climate since on long time scales the large ice sheets might feed back on global temperature levels. We revised our manuscript according to that carefully.

We added a discussion about the realism of our results and find that they are in accordance with previous efforts (Lunt et al., 2012). For this, we performed a range of additional simulations (see below, ll. 162 in the manuscript and supplementary Fig. S4).

Major comments

M1: Several specific cryosphere components are chosen based on the perception that they are “at risk of transgressing a critical threshold which could cause large-scale, possibly irreversible changes”. The authors then explore the global temperature response to the removal of each cryosphere component individually and together and claim that this additional warming could have bearing on the Paris targets. I believe this premise to be flawed in several ways.

Firstly, not all of the cryosphere components chosen are at risk of passing a critical threshold or tipping point beyond which loss would be irreversible. Summer Arctic sea ice loss in particular is thought to be reversible with respect to global temperature (e.g., doi: 10.1175/JCLI-D-14-00654.1, 10.5194/tcd-5-2349-2011, 10.1029/2011GL048739, doi:10.1029/2010GL045698, 10.1038/nature09653). Moreover, the contribution of sea ice loss to global temperature is already included in the Earth System Model simulations used to quantify the relationship between cumulative carbon and global temperature and thus is already included in the Paris targets. By artificially removing sea ice the authors seem to be implicitly making the argument that GCMs/ESMs are missing a mechanism by which sea ice is lost irreversibly at low warming levels (e.g., 1.5K), but they do not describe or justify such a mechanism thus making the choice seem quite arbitrary. One could just as easily induce the loss of all low clouds or all snow within a model at 1.5K and quantify the additional warming that would ensue, but such experiments would not be meaningful or have any bearing on temperature targets.

It is true that the Arctic summer sea ice loss is reversible and we clearly did not intend to claim that it is not. We are now stating that more clearly in the manuscript with the help of the publications that you mentioned above - thank you. (see ll. 82 ff.)

We are also aware that current GCMs already include the loss of the sea-ice in their simulations concerning the Paris Agreement which we write in lines 53 - 60 now.

However, in the GMC simulations that you mentioned it appears that the meltdown of the Arctic summer sea ice is underestimated (even for RCP 8.5), see Figure below. Therefore, and since the Arctic summer sea ice is one of the main cryosphere elements, we think that it is worthwhile to evaluate its potential additional warming.

(adapted from Stroeve et al., 2012; compare also to Overland et al., 2011)

The loss of mountain glaciers is also, to my knowledge, a reversible process. This experiment seems better justified by the fact that much of the glacier loss will happen this century yet may not be included in many GCMs/ESMs and thus is a potential source of additional warming that has relevance for temperature targets. If this is the first study to quantify the global surface albedo feedback and additional warming associated with mountain glaciers then this should be noted. I suspect that it is not, in which case results should be compared to previous efforts.

The loss of the mountain glaciers is only reversible when the melt-elevation feedback (see for example in Levermann & Winkelmann (2016)) does not play a major role. This is the case for most low-height glaciers, but might play a role for glaciers of larger vertical size as for example in the Himalaya. But certainly, it can better be understood when we frame the loss of the mountain glaciers in terms of their neglectance in many GCMs as you advised. (see II. 84 ff.)

There are many studies that compute the commitment to sea level rise from mountain glaciers, but we are not aware of any study that computes the temperature commitment.

The complete loss of the Greenland and West Antarctic Ice Sheets may be irreversible, but the timescales for ice sheet loss are centuries to millennia meaning that there is little relevance for the Paris temperature targets. It has been known for some time that the loss of ice sheets would induce additional warming, with estimates of the Earth System Sensitivity (ESS) ranging from about 30% to 100% above the Equilibrium Climate Sensitivity (ECS) which excludes slow feedback changes (e.g., doi: 10.1038/ngeo706, 10.2174/1874282300802010217, 10.1038/nature11574). I was struck by how small the additional warming from GIS and WAIS were (0.14K and 0.06K, respectively, on top of the 1.5K baseline), and found the manuscript lacking in its discussion of these results in the context of the many previous studies.

It is known that Earth System Sensitivity can be substantially larger than climate sensitivity, but there are many more processes contributing to that besides loss of ice sheets (e.g. Fig. 1 in the PALEOSENS paper). In particular the contribution of vegetation is important. There are a few papers that computed the change in temperature as a result of ice sheet loss, e.g. Lunt et al., 2012 (Fig. 4e), Lord et al., 2017 (Fig. 4a).

Both studies show a pronounced warming in the proximity of the locations where ice is removed, which is in good agreement with our results.

However, both studies used an ice sheet reconstruction for the late Pliocene (PRISM), which includes the loss of most of Greenland and West Antarctica, but also of substantial parts of East Antarctica. With these prescribed ice sheets Lunt et al. (2012) found a global warming of 0.7°C , which is more than our warming of 0.2°C , but the discrepancy comes mainly from East Antarctica, which is still intact in our simulations. For comparison, we show here below (and in supplementary Fig. S4) a comparison, where we remove all the cryosphere elements (WAIS, GIS, Arctic summer sea ice, mountain glaciers and EAIS) including the whole Antarctic Ice Sheet. We then end up with a temperature difference of 0.82°C , where we used a CO_2 concentration of 280 ppm. This is in accordance with the 0.7°C found in Lunt et al. (2012). We added some of this discussion in the revised version of the manuscript. (see II. 163 ff. and supp. Fig. S4)

Left: Our experiment, where we removed all cryosphere elements at 280 ppm. We end up with a warming of 0.82°C , **Right:** Lunt et al., 2012 (Fig. 4e): Warming of 0.7°C (see Table 1).

In general, the 1.5K to 2K Paris temperature target has been chosen to avoid the loss of the GIS and WAIS. If the authors then want to claim that their results have bearing on these temperature targets then they have to provide some evidence that that the irreversible loss of these cryosphere components will occur at temperatures lower than 1.5K to 2K, but I see no

such discussion. Overall this study has the feel of a modelling exercise with little to no connection to reality.

Yes, the Paris temperature has been chosen to avoid a high probability of tipping of GIS and WAIS. However, following Schellhuber, et al. (2016), there is a non-negligible probability fraction that GIS, WAIS, the mountain and the Arctic summer sea ice might be lost even at these intermediate climate warming scenarios below Paris. For example the critical temperature range for the Greenland Ice Sheet has been found to range between 0.8-3.2°C of global warming computed with the Ice Sheet model SICOPOLIS (see Robinson, et al.(2012)). (see II. 8f.)

Following this argument, there is the risk of transgressing the 1.5 or the 2.0°C temperature threshold on long time scales even if it will be achieved until 2100 and without the combustion of further fossil fuels.

M2: The accuracy of the results (additional warming commitment) depends critically on the feedbacks as represented in CLIMBER-2. I recognize that the authors have put much time and effort into spanning a range of cloud and lapse rate feedback parameters to represent a range of ECS values (1.5-4.5 K). But I see some technical issues that need to be addressed or explained.

- Key to the results is the strength of the feedbacks induced by the cryosphere changes. The magnitude of the LR+WV and cloud feedbacks (Table 1) seem reasonable compared to GCMs, theory and observations. However, the albedo feedback seems far too large with values ranging from 1.7 W/m²/K (loss of GIS) to 2.4 W/m²/K (loss of summer Arctic sea ice). To put this in perspective, the surface albedo feedback in GCMs under abrupt CO₂ quadrupling, where most summer Arctic sea ice is lost, is typically about 0.4 W/m²/K. Table 1 also shows the sum of all feedbacks to generally exceed 3.2 W/m²/K, which is the value of the Planck feedback and thus the maximum allowed feedback value to produce a stable climate. Clearly either something has gone wrong in these feedback calculations or CLIMBER-2 has a very strange climate. The authors should examine the surface albedo feedback in the baseline simulations (without induced cryosphere loss) to see if it is at all realistic and reexamine their Calculations.

We agree with the reviewer that the way the feedbacks were represented in Table 1 was a bit misleading. Following also a similar comment by reviewer #2 we decided to change the way the numbers in Table 1 are presented.

For the calibration runs we did a standard feedback analysis for CO₂ doubling experiments using the partial radiation perturbation method to compute the feedback factors (e.g. Wetherald and Manabe, 1988). These feedback factors can be directly compared to results of CMIP models and we use this as one of our constraints for the model ensemble (Fig. S1a). For the calibration all 4 cryoelements are prescribed as present.

The numbers in Table 1 were instead derived from a perturbed run with cryosphere elements removed (each separately and all together), meaning that we computed the feedback factors from the radiation perturbation at TOA when the cryosphere elements were removed.

We agree with the reviewer that the numbers presented in Table 1 could be misleading, because what is usually reported are the numbers of the feedback factors for a CO₂ doubling. To avoid confusion, in Table 1 we now report the radiative forcing introduced by the removal of the cryosphere components through changes in albedo, water vapor, lapse rate and clouds. (see II. 185 ff. and Tab. 1)

In the revised Methods section we also describe in more detail how the feedback analysis is performed.

- The very small range of uncertainty for the global warming response to loss of Earth's cryosphere (0.37 +/- 0.03 K) is simply not believable. How can this small range be reconciled with the factor of three range of ECS spanned by the model (1.5-4.5 K), for instance?

Yes, you are right and we also see that this point needs better explanation and representation in order to avoid confusion. This is why we do not show standard deviations anymore in our temperature plots (and in the main text), but we refer to the whole ensemble now which shows the larger spread of our results. We use this procedure now as we think that no ensemble member should be treated special since all of them passed the calibration process.

For example the difference from the minimal value to the maximal value is larger (which is hidden by a simple standard deviation), for example at a CO₂ concentration of 400 ppm, the smallest ensemble member gives a warming of 0.36°C, the largest 0.51°C, for the removal of all four cryosphere elements (mean 0.42°C).

All in all, we think that the headline result of now 0.42°C (0.36 - 0.51°C) is reasonable (beforehand 0.37±0.03°C). We provide further explanation in response to your points raised in terms of the additional warming of WAIS and GIS above. (see all results, figures and especially II. 156 ff.)

Minor comments

- L29-31: I encourage the authors to define what they mean if they are to use the term 'tipping element'. Are the changes of these cryosphere components reversible with respect to global temperature or not, and how do we know? This seems key to motivating this line of work (see above).

Here, we are using the definition of tipping elements following Lenton, et al. (2008). This definition does not require a tipping element to be irreversible such that for example the Arctic summer sea ice and the mountain glaciers can be counted as tipping elements as

well. Because that seems to have been the source of confusion, we now stated this explicitly in the introduction paragraph.

Tipping elements are irreversible as long as there is a feedback process that locks-in a certain element in a certain state. For the WAIS and the GIS (and potentially also for the mountain glaciers), this is the melt-elevation feedback (Levermann & Winkelmann (2016)). This feedback kicks in at a certain critical temperature when the upper part of the Ice Sheets start melting.

The Arctic summer sea ice is reversible since there is no such feedback process. The excess heat that warms up the ocean in summer after the complete loss of sea ice, is lost to space during winter when new winter sea ice emerges. (ll. 38 ff. and 82 ff.)

- L173-174: I do not understand this sentence. There are several factors contributing to positive cloud feedback in GCMs, not just associated with cloud amount and not just in mid to high latitudes (see for example doi: 10.1038/nclimate3402).

Thanks for pointing to this paper. We changed the respective sentence in the manuscript. (see ll. 205 f.)

- L185-187: As noted above, Arctic sea ice loss is already included in ESMs used to quantify the maximally allowed CO2 emissions.

We rephrased this sentence emphasizing that this warming is not due to the combustion of CO2, but we are not referring to the Paris range anymore. (see ll. 224 ff.)

- L225-227: Is the ice sheet albedo replaced with the lbedo of ocean, bare land, or land with vegetation? Is isostatic rebound accounted for in the response to loss of GIS or WAIS?

The ice sheet albedo is replaced by the albedo of bare land, but can evolve freely into any kind of vegetation, bare land or snow depending on the land surface cover that is simulated interactively by the CLIMBER-2 vegetation module. Thus, our results can be expected to be robust if we change the albedo to the albedo of bare land or vegetation. In case of the marine regions of the West Antarctica, the albedo is replaced by ocean albedo, but can then evolve freely, i.e., mostly into sea ice regions. Isostatic rebound is neglected. We these comments in our manuscript. (see ll. 267 ff.)

- L276-279: This behavior in GCMs usually has to do with deep convective variability in the region of the Weddell sea polynya. Is this what you mean by 'the same behavior' in your model. I doubt it. I suspect it is a model artefact associated with your idealized zonal average ocean.

In our new improved calibration, this behavior does not occur anymore for the investigated CO2 range, so we leave these sentences out.

Reviewers' comments:

Reviewer #1 (Remarks to the Author):

Dear editors, dear authors,

Thanks for very carefully considering my comments, in a meaningful and constructive spirit.

After my second reading of the manuscript and, I realize there are two caveats that limit the scope of the work.

1) The extra-warming, in particular by sea ice is somehow already included in ESM projections, in particular in the context of the upcoming CMIP6, where trends in sea ice are somehow more in line with observations. Results are unpublished yet, but were shown by several groups at international conferences.

2) Freshwater released from ice sheet melting is ignored from the experiments, whereas it is known to matter especially, in the short term. For instance, Fichefet et al (GRL2003) have shown a substantial regional cooling following the release of freshwater from the melt of the Greenland Ice Sheet in the 21st century. Swingedouw et al (GRL 2008) show that the melting of the Antarctic ice sheet increases stratification in the Southern Ocean and dampens the effects of the ice albedo feedback. Hence, the figures presented in the paper neglect the dampening contribution of freshwater release and may appear to overestimate climate response to the loss of continental ice.

In my current understanding of the paper, these two items question the relevance of the paper's results into the Paris agreement debate.

I still think the paper is a very useful contribution, if it is made clear if these two caveats are properly addressed in the manuscript.

Best wishes,

Martin Vancoppenolle

Reviewer #2 (Remarks to the Author):

My first review had already agreed that the manuscript was a useful contribution to modelling efforts helping to understand and estimate climate sensitivity. The clarifications of the revision help sort out my confusion (if I understood the clarification correctly, see below), but make me slightly less enthusiastic about the "surprise value" of the results.

The revision addresses all my minor concerns. My main concern was the proper definition of the feedback factors in Table 1 and an explanation of their low level of uncertainty. The presentation of these numbers has been clarified in the revision. However, the clarification in the response and the clarification in the manuscript still suggest two different interpretations. So, this is an issue that may require a slightly more explicit statement. A sentence explaining their low level of uncertainty is also still missing.

The apparent contradiction:

The manuscript states in the section "Methods->Model calibration and computation of feedbacks" (l1288-291):

"Each of our 39 ensemble members ... is constructed from a pair of simulations: one control run at 280 ppm and one perturbed run at a CO₂ doubling of 560 ppm. We then compute the magnitude of the fast climate feedbacks between these pairs of runs (see supp. Fig. S1 a)."

This appears to imply that the perturbation involves doubling of CO₂. The text refers to Fig S1a, but since this text is under the heading "computation of feedbacks" I initially assume(d) that this applies to all quantities called "feedback" in the paper, including those in Tab.1.

Contradicting this, in ll185-190, the manuscript states "Here, we give the strength of the feedbacks in change of radiative forcing in [W/m²] instead of feedback strength per degree of warming in [W/m²/K] to avoid confusion with the usual feedback computation that are computed at a doubling of CO₂ since we compute the feedbacks here for the same carbon dioxide concentration (400 ppm) between a control and a perturbed run."

So, what is the precise perturbation? I was originally under the impression that the table showed the change of the CO₂-doubling feedback factor when removing the respective cryosphere element.

Response and text now appear to suggest that the numbers in Tab.1. are not feedback factors at all. The numbers show change in radiative forcing [W/m²] when removing a cryosphere element and how the change in radiative forcing is divided between albedo, water vapor/lapse rate and clouds. The "feedbacks" in Tab.1 have the unit W/m² which indicates that the word feedback is misleading, unless it is linked back to further cryosphere change.

Contrast this with the classical case for (eg) the "albedo feedback factor" for CO₂ doubling (as I understand it from Fig S1): radiation amount x trapped by CO₂ doubling causes heating, which causes ice melt, which causes additional radiation amount y trapped by darker surface. In short, more radiation trapped x causing more radiation trapped y = positive feedback strength y/x (no dimension). In principle feedback factors should be dimensionless. The unit W/m²/K in Fig S1 is ok-ish since power density [W/m²] and temperature change [K] are closely related.

So, the authors should refer to the columns of Tab.1 as (eg) radiation trapping mechanisms. In its current form the notion of "feedback" is misleading as it gives the impression that the removal of cryosphere elements makes the climate more prone to (eg) an albedo-feedback induced instability. The opposite is true, since the removal of the cryosphere element reduces the area where reflection changes the most. May-be, among climate researchers the radiation trapping mechanisms are always referred to as feedbacks, but the general audience of Nature Communications might be as confused as I was by the original manuscript. I had found the change of feedbacks (assuming they were amplifiers of CO₂ doubling effects) very remarkable, even inexplicable. I had rated this as the most interesting result of the manuscript, since the debate about realistic feedback factors is still open (see figure S1).

Reviewer #3 (Remarks to the Author):

Second review of "Additional global warming commitment due to crossing critical thresholds within the Earth's cryosphere" submitted to Nature Communications by Nico Wunderling, Matteo Willeit, Jonathan F. Donges, and Ricarda Winkelmann.

The authors have made many changes to the manuscript but have not addressed my most substantial concerns. I reiterate these concerns below and provide additional comments on the new elements that have been added to the analysis. My assessment is that this work still needs substantial revision in a number of ways if it is to become publishable. Given that little progress has been made during the last round of revision my recommendation is that it be rejected so that

the authors can have the time to consider the reviews and make the necessary revisions before resubmission.

Major comments

M1: One of my previous major concerns was the framing of the work in terms of critical cryosphere thresholds beyond which we would be committed to additional warming. In response, the authors modified the introductory text to emphasize the concept of a "tipping element" which no longer requires that a system have an actual tipping point or irreversible behavior but instead only requires that "an element quickly degrades at a certain critical parameter" value.

I find these revisions lacking in multiple ways. The use of the "tipping element" terminology is both confusing and unnecessary. It is confusing because it is so vague a definition that it will lead readers to incorrectly believe that the cryosphere components were chosen based on their risk of irreversible collapse, even while some components (sea ice and possibly glaciers) are not thought to exhibit tipping point behavior. It is also confusing because the one cryosphere component that likely will show tipping point behavior, land ice sheets, will not "degrade quickly" but instead will take centuries to millennia to be lost even if a critical threshold is passed.

The "tipping element" terminology is unnecessary because the work explores the climate response to the complete removal of cryosphere components rather than providing an assessment of the likelihood of irreversible behavior or the timescale on which the loss of the cryosphere would occur. That is, the work is far better justified by stating that it is possible that various cryosphere components will be lost at lower temperatures than commonly thought and thus this study examines the global climate response to those hypothetical collapses.

I would like to see the authors to remove their discussion of tipping elements entirely. This would amount to a major revision to the introduction.

M2: The authors have changed the way in which they report the range of their climate responses to cryosphere collapse. While previously they reported one-standard-deviation ranges, which were extremely small, they now report the full range of ensemble members which is quite a bit larger. However, I don't think this is the correct approach as it only relies on the outlier models and gives no sense of the statistics of the model spread. I suggest that the authors consider using alternative measures for the spread, such as the interquartile range.

More importantly, I am unable to reconcile the previously-reported small range of warming uncertainty across ensemble members (standard deviation of 0.03K on top of an extra warming of 0.4K) with the much larger range now reported and shown in Figure S3. What has changed in the methods to cause the large increase in uncertainty, and why was it not discussed in the response to reviews? Moreover, Figure S3 seems to show a considerably larger standard deviation range (about 0.5K) than suggested in the text, and seems to show many ensemble members with anomalous cooling under removal of cryosphere components. This is confusing and needs explanation before I can make any assessment of the robustness of the results.

M3: It appears to me that the authors are trying hard to shape this work into newsworthy soundbites, but that it may be better suited in a more specialized journal where the results can be described in greater detail and the findings not oversold. An example of this overselling is the use of the "tipping element" framing, as discussed above. Another is the concluding paragraph of the manuscript which states:

"This implies that climate change and global warming would be self-amplifying in such scenarios and leaves the question if the direct interaction between tipping elements combined with the mean-field effect on GMT exerts positive feedbacks that are strong enough to trigger cascading transitions. This would, in turn, increase the risk of the Earth system moving towards a hothouse

state.”

This statement is unjustified by the results of the work. There is no evidence presented that the different cryosphere components would strongly interact, as implied, as the study only considers the complete removal of each component or all at once and does not provide any assessment of the temperature at which a critical threshold would occur or, indeed, if any actually exist. It seems unlikely to me that the loss of one cryosphere component would be the sole trigger for the loss of another. And the concluding thought regarding moving the system towards a hothouse state is entirely unjustified. The results show, if anything, relatively small additional warming from the loss of substantial cryosphere components that do not move the system to anywhere near a runaway greenhouse effect or a hothouse climate.

I urge the authors to read through the entire text to remove statements like this that are not supported by the results and to stick to reporting what has been done without the spin.

M4: It is implied in many places that the results have implications for the Paris targets and for warming over the 21st century. However, the results strictly apply only at equilibrium, which takes multiple millennia to be reached. This should be clarified in the abstract and elsewhere.

If implications for the Paris agreement are to be discussed, it should also be noted (on L112-114 and elsewhere) that the freshwater input from the loss of cryosphere components has the potential to completely cancel the warming you estimate on century timescales. For instance, recent papers estimate a global cooling of about 0.4K by 2100 from the freshwater input associated with loss of West Antarctic Ice Sheet (doi: 10.1038/s41586-019-0889-9, 10.1038/s41586-018-0712-z), which would more than offset the warming found here.

Minor comments:

- L5-9: Right away you are implying that the cryosphere components you consider are those with critical thresholds that could be crossed, but that is incorrect (to our knowledge) in the case of summer sea ice and not supported here for the other components. Please reframe as discussed above.

- L16 and L23: What are these numbers? Median or mean? Standard deviation or full ensemble spread?

- L23 and elsewhere: Clarify that these values are specific to a given CO₂ level and background warming. If CO₂ is quadrupled and all Arctic summer sea ice is lost, then the loss of summer sea ice would cause no additional warming, right?

- L56: I don't understand the use of the word "integrate" in this context.

- L79 and L147-148: The Antarctic Ice Sheet is an example of something that may truly have a tipping point but would not degrade quickly, thus not satisfying the "tipping element" definition used here.

- L147-148: How does this have any relevance for the Paris agreement, which aims to hold warming to less than 1.5K or 2K on decadal to centennial timescales? On timescales of millennia, other effects such as the long-term drawdown of CO₂ by the oceans would become important, so it's not clear how these results would be relevant for temperature targets out that far.

- L157-159: This is the first place that you explain what the values in brackets represent. You also still have not said what the main estimate represents. Is it a median or mean, for instance?

- L187-190 and Table 1: I do not understand why you have chosen to report values in W/m²

rather than $W/m^2/K$ as before. The issue was not that reporting the values as a feedback was confusing or misleading, but instead that your values made no sense. Your response to reviewers did not explain what had gone wrong so I am unable to assess whether the issue has been corrected or swept under the rug by a change in units. I think framing this in terms of feedbacks ($W/m^2/K$) as done previously is much more natural since the additional W/m^2 from the loss of the cryosphere components is difficult to compare with the other feedbacks acting.

- L214: "exemplary" is the wrong word and could be removed.

- L231-232: This does not follow from the results. This effect needs to be taken into account only if these cryosphere components are actually lost at the temperature values explored here. Yet no evidence is given that they will be.

Dear Reviewers,

Thank you very much for your insightful comments on our manuscript. We are grateful for the opportunity to further improve our paper in light of your comments. We carefully revised our manuscript with the requested major revisions and are confident that they will help to clarify the open points raised.

Major changes now include:

1. **Framing**: We reworked our paragraphs and toned down the parts about tipping points as requested: This includes changes in the title, abstract, conclusion and major changes in the introduction and conclusion.
2. **Statistics**: We changed the error bars that we report from full ensemble spread to interquartile ranges (box whiskers plots) to give a better impression of the distribution of our ensemble.
3. **Feedbacks**: We worked on resolving the misunderstanding that arose in the feedbacks part of our manuscript, clarified open points and discussed the small spread of our results here and elsewhere in the manuscript carefully. Furthermore, we compare the additional radiative forcing we find from the removal of some cryosphere components to values reported literature.

Besides the major revisions, please see below for a point-by-point response to your comments. You can find the colour-marked changes in blue in our manuscript.

We are grateful for this opportunity to further improve our manuscript and are convinced that the reviewers' comments have been addressed in line with the high standards of Nature Communications.

Sincerely yours,

Nico Wunderling, Matteo Willeit, Jonathan F. Donges & Ricarda Winkelmann

Reviewer #1 (Remarks to the Author):

Dear editors, dear authors,

Thanks for very carefully considering my comments, in a meaningful and constructive spirit.

After my second reading of the manuscript and, I realize there are two caveats that limit the scope of the work.

1) The extra-warming, in particular by sea ice is somehow already included in ESM projections, in particular in the context of the upcoming CMIP6, where trends in sea ice are somehow more in line with observations. Results are unpublished yet, but were shown by several groups at international conferences.

Yes, this is true, sea-ice was already included in CMIP6 and CMIP5 models (see eg Niederdrenk et al., Geophysical Research Letters (2018) and Notz et al., Journal of Advances in Modeling Earth Systems (2013)). We make reference to this now more explicitly in our manuscript (ll 36-38 and 71-73).

2) Freshwater released from ice sheet melting is ignored from the experiments, whereas it is known to matter especially, in the short term. For instance, Fichefet et al (GRL2003) have shown a substantial regional cooling following the release of freshwater from the melt of the Greenland Ice Sheet in the 21st century. Swingedouw et al (GRL 2008) show that the melting of the Antarctic ice sheet increases stratification in the Southern Ocean and dampens the effects of the ice albedo feedback. Hence, the figures presented in the paper neglect the dampening contribution of freshwater release and may appear to overestimate climate response to the loss of continental ice.

Thank you very much for pointing that out. This is of course true, however, we intended to focus on the long-term equilibrium response of the climate system in terms of temperature increase with regard to the removal of cryosphere elements. Following your recommendation, we included this caveat clearly in our manuscript with the respective references that you mentioned (ll 101-105).

In my current understanding of the paper, these two items question the relevance of the paper's results into the Paris agreement debate.

We reframed the part about the relevance for the Paris agreement since we agree that the temperature feedbacks for the Arctic summer sea ice are already included in CMIP-5 simulations and match better with observations in the new generation of CMIP-6 simulations. For the other elements (Mountain glaciers, West Antarctic Ice Sheet, Greenland Ice Sheet) investigated in our study, the time horizon of disintegration is

clearly beyond 2100 with only smaller contributions beforehand. So we removed statements about the Paris agreement from our manuscript. Thank you for this advice.

I still think the paper is a very useful contribution, if it is made clear if these two caveats are properly addressed in the manuscript.

Best wishes,

Martin Vancoppenolle

Reviewer #2 (Remarks to the Author):

My first review had already agreed that the manuscript was a useful contribution to modelling efforts helping to understand and estimate climate sensitivity. The clarifications of the revision help sort out my confusion (if I understood the clarification correctly, see below), but make me slightly less enthusiastic about the "surprise value" of the results.

The revision addresses all my minor concerns. My main concern was the proper definition of the feedback factors in Table 1 and an explanation of their low level of uncertainty. The presentation of these numbers has been clarified in the revision. However, the clarification in the response and the clarification in the manuscript still suggest two different interpretations. So, this is an issue that may require a slightly more explicit statement. A sentence explaining their low level of uncertainty is also still missing.

The apparent contradiction:

The manuscript states in the section "Methods->Model calibration and computation of feedbacks" (l1288-291):

"Each of our 39 ensemble members ... is constructed from a pair of simulations: one control run at 280 ppm and one perturbed run at a CO₂ doubling of 560 ppm. We then compute the magnitude of the fast climate feedbacks between these pairs of runs (see supp. Fig. S1 a)."

This appears to imply that the perturbation involves doubling of CO₂. The text refers to Fig S1a, but since this text is under the heading "computation of feedbacks" I initially assume(d) that this applies to all quantities called "feedback" in the paper, including those in Tab.1.

Contradicting this, in l1185-190, the manuscript states "Here, we give the strength of the feedbacks in change of radiative forcing in [W/m²] instead of feedback strength per degree of warming in [W/m²/K] to avoid confusion with the usual feedback computation that are computed at a doubling of CO₂ since we compute the feedbacks here for the same carbon dioxide concentration (400 ppm) between a control and a perturbed run."

So, what is the precise perturbation? I was originally under the impression that the table showed the change of the CO₂-doubling feedback factor when removing the respective cryosphere element.

Response and text now appear to suggest that the numbers in Tab.1. are not feedback factors at all. The numbers show change in radiative forcing [W/m²] when removing a cryosphere element and how the change in radiative forcing is divided between albedo, water vapor/lapse rate and clouds. The "feedbacks" in Tab.1 have the unit W/m² which indicates that the word feedback is misleading, unless it is linked back to further cryosphere change.

Contrast this with the classical case for (eg) the "albedo feedback factor" for CO₂ doubling (as I understand it from Fig S1): radiation amount x trapped by CO₂ doubling causes heating, which causes ice melt, which causes additional radiation amount y trapped by darker surface. In short, more radiation trapped x causing more radiation trapped y = positive feedback strength y/x (no dimension). In principle feedback factors should be dimensionless. The unit W/m²/K in Fig S1 is ok-ish since power density [W/m²] and temperature change [K] are closely related.

So, the authors should refer to the columns of Tab.1 as (eg) radiation trapping mechanisms. In its current form the notion of "feedback" is misleading as it gives the impression that the removal of cryosphere elements makes the climate more prone to (eg) an albedo-feedback induced instability. The opposite is true, since the removal of the cryosphere element reduces the area where reflection changes the most. May-be, among climate researchers the radiation trapping mechanisms are always referred to as feedbacks, but the general audience of Nature Communications might be as confused as I was by the original manuscript. I had found the change of feedbacks (assuming they were amplifiers of CO₂ doubling effects) very remarkable, even inexplicable. I had rated this as the most interesting result of the manuscript, since the debate about realistic feedback factors is still open (see figure S1).

Feedback discussion:

We agree that our notion of feedbacks was confusing since we are computing two different model setups that are coupled back to the original system which is why we called both feedbacks.

First: The usual notion of feedbacks is used in our calibration procedure (Fig. S1) that is applied in the 2xCO₂ experiments which were used as a filtering method to select appropriate ensemble members. With the 39 ensemble members that we select all further simulations are performed and reported in the main manuscript and in the supplement.

Second: The values that are reported in Tab. 1 are obtained from a different kind of experiment. Here, we actually compute the additional radiative forcing at the top of the atmosphere that originates from the disappearance of the cryosphere elements at the same atmospheric CO₂ concentration. In Tab. 1 we report the values for disintegration of the cryosphere elements at 400 ppm.

Similar investigations have been performed for the removal of Arctic sea ice. For a removal of one month during summer an additional radiative forcing of 0.3 W/m² is reported (Hudson, 2011, Journal of Geophysical Research: Atmospheres) which is in good agreement with Flanner et al. (2011, Nature Geoscience). We find a higher additional radiative forcing of 0.41+0.08 W/m² = 0.49 W/m² (Tab. 1: Albedo + Clouds). This value is higher since we have low sea ice for approximately five months in our perturbed experiments instead of one as in Hudson, 2011 (II 208-214).

Thank you very much for helping us to resolve this misunderstanding about the proper usage of the term “feedback” here. Now we very much agree that what is reported in Tab. 1 could be confusing if called a feedback factor. Instead we are now referring to that as “additional radiative forcing” in the manuscript. Please find our major changes in **II. 175-224**.

It appears that different communities handle the notion of the word *feedbacks* in a different way and we are confident that our manuscript is easier to grasp and understand now.

Low level of uncertainty:

The low level of uncertainty observed in our modeling results can be explained as follows: the climate sensitivity for our calibration measurements lies in a reasonable range as observed in CMIP4 and 5 simulations (see black dots in Fig. S1b), namely between 2.0 - 3.75°C. From these calibration measurements we have generated 39 ensemble members during our calibration procedure.

For each of these ensemble members the following experiment is performed for creating the values in Tab. 1: we compute additional radiative forcing between an experiment with disintegrated cryosphere element(s) and compare this to the experiment with intact cryosphere at the same atmospheric CO₂ concentration (400 ppm). This means that only the difference between these two experiments is reported in Tab. 1, and this difference does not vary much between different members of the ensemble. We added a description of that to our manuscript (**see II. 329-334**).

For the same reason, the error bars in Fig. 3 (see also explanatory figure below at 400 ppm for the removal of all investigated cryosphere elements at the same time) for instance are also relatively small, because we plot the results from the difference (red) and no absolute temperature values (green: disintegrated cryosphere elements; black: intact cryosphere elements) between disintegrated and intact cryosphere. In the plot, two y-axes are plotted where intact and disintegrated cryosphere belong to the left and the difference to the right y-axis. This means that our CLIMBER-2 ensemble is robust against the same perturbations in the cryosphere components which we constructed aiming at covering a range of sensitivities and different strengths of the feedbacks by the variation of the parameters in Tab. S1.

Please note that we now show median values as well as the interquartile range and the full ensemble spread in the manuscript in box whiskers plots, where applicable.

Reviewer #3 (Remarks to the Author):

Second review of “Additional global warming commitment due to crossing critical thresholds within the Earth’s cryosphere” submitted to Nature Communications by Nico Wunderling, Matteo Willeit, Jonathan F. Donges, and Ricarda Winkelmann.

The authors have made many changes to the manuscript but have not addressed my most substantial concerns. I reiterate these concerns below and provide additional comments on the new elements that have been added to the analysis. My assessment is that this work still needs substantial revision in a number of ways if it is to become publishable. Given that little progress has been made during the last round of revision my recommendation is that it be rejected so that the authors can have the time to consider the reviews and make the necessary revisions before resubmission.

Major comments

M1: One of my previous major concerns was the framing of the work in terms of critical cryosphere thresholds beyond which we would be committed to additional warming. In response, the authors modified the introductory text to emphasize the concept of a “tipping element” which no longer requires that a system have an actual tipping point or irreversible behavior but instead only requires that “an element quickly degrades at a certain critical parameter” value.

I find these revisions lacking in multiple ways. The use of the “tipping element” terminology is both confusing and unnecessary. It is confusing because it is so vague a definition that it will lead readers to incorrectly believe that the cryosphere components were chosen based on their risk of irreversible collapse, even while some components (sea ice and possibly glaciers) are not thought to exhibit tipping point behavior. It is also confusing because the one cryosphere component that likely will show tipping point behavior, land ice sheets, will not “degrade quickly” but instead will take centuries to millennia to be lost even if a critical threshold is passed.

The “tipping element” terminology is unnecessary because the work explores the climate response to the complete removal of cryosphere components rather than providing an assessment of the likelihood of irreversible behavior or the timescale on which the loss of the cryosphere would occur. That is, the work is far better justified by stating that it is possible that various cryosphere components will be lost at lower temperatures than commonly thought and thus this study examines the global climate response to those hypothetical collapses.

I would like to see the authors to remove their discussion of tipping elements entirely. This would amount to a major revision to the introduction.

We agree with the referee that the tipping elements framing is actually not required to motivate the study and to discuss its results. We rewrote our introduction including the

major changes requested. We removed the statements on tipping elements completely from the introductory part by starting with a review of the recent developments of the investigated cryosphere components (II 28-64) and then stating that we intend to assess the additional global warming caused by their complete disintegration (II 65-77).

During the first round of revisions we had the impression that our notion of tipping elements was not explained in an understandable way such that we were elaborating on this, but apparently we did not understand the way your comment was intended. We apologize for that and hope we were able to clarify this point remaining from the first round of revisions.

Of course we also removed the tipping elements framing from the conclusion and the abstract as well (II 4-27 and 225-243).

M2: The authors have changed the way in which they report the range of their climate responses to cryosphere collapse. While previously they reported one-standard-deviation ranges, which were extremely small, they now report the full range of ensemble members which is quite a bit larger. However, I don't think this is the correct approach as it only relies on the outlier models and gives no sense of the statistics of the model spread. I suggest that the authors consider using alternative measures for the spread, such as the interquartile range.

We used the full range of ensemble spread since we already calibrated our ensemble with CMIP feedback constraints leaving us with the impression that no member of the ensemble should be better than another. But we agree with the reviewer that reporting the whole ensemble range does not give any information on the distribution of the ensemble members. We therefore decided to report both, the whole ensemble range and the interquartile range (and the median) in the figures in the revised manuscript to provide a more complete picture of the model spread (see II 13-14, II 115-116, Figs. 3, 4, S2, S3).

More importantly, I am unable to reconcile the previously-reported small range of warming uncertainty across ensemble members (standard deviation of 0.03K on top of an extra warming of 0.4K) with the much larger range now reported and shown in Figure S3. What has changed in the methods to cause the large increase in uncertainty, and why was it not discussed in the response to reviews?

The median additional warming that we find for the case of removal of the West Antarctic Ice sheet, the Greenland Ice sheet, the mountain glaciers and the Arctic summer sea ice is 0.43°C (interquartile range: 0.39-0.46°C, full ensemble spread: 0.36-0.51°C). This is the value that we show in Fig. 3 at 400 ppm, where 400 ppm equals a global mean temperature increase of 1.5°C in CLIMBER-2. We agree that the interquartile ranges

(whose uncertainty range is comparable with the standard deviation reported earlier) are of relatively low spread. Please see below for a more detailed discussion.

Low level of uncertainty:

The low level of uncertainty can be explained as follows: the climate sensitivity for our calibration measurements is in a reasonable range as observed in CMIP4 and CMIP5 simulations (see black dots in Fig. S1b), namely between 2.0 - 3.75°C. From these calibration measurements we have generated 39 ensemble members during our calibration procedure. Thus, the absolute temperature difference between the ensemble members is considerably large.

For each of these ensemble members the following experiment is performed for creating the values in Fig. 3: we compute the temperature at a certain fixed CO₂ concentration with intact and disintegrated cryosphere which gives us two values with larger uncertainty. Then, the difference between these two values for each of the 39 ensemble pairs is plotted in Figure 3 for each CO₂ concentration as box whiskers plots.

The explanatory figure below shows the spread of the experiments with disintegrated (green) and intact cryosphere elements (black) for the removal of all investigated cryosphere components at the same time for an atmospheric CO₂ concentration of 400 ppm. The red dots show the difference for each of the 39 green and black pairs. The plot uses two y-axes, where the left y-axis is for green and black dots and the right y-axis for red dots. This demonstrates that the spread of the red dots is remarkably smaller than for the green and black values. This means that our CLIMBER-2 ensemble is robust against the same perturbations in the cryosphere components which we constructed aiming at covering a range of sensitivities and different strengths of the feedbacks by the variation of the parameters in Tab. S1.

The same argument for small error bars is valid for the additional radiative forcing values reported in Tab. 1 as well as for Fig. S3.. We added a statement to the paper in II. 329-334.

Moreover, Figure S3 seems to show a considerably larger standard deviation range (about 0.5K) than suggested in the text, and seems to show many ensemble members with anomalous cooling under removal of cryosphere components. This is confusing and needs explanation before I can make any assessment of the robustness of the results.

Figure S3b:

The ensemble spread in our old Fig. S3a was correct, but there was a small mistake in drawing the error bar of Fig. S3b in my plotscript. Thank you very much for calling my attention to this, of course it would have made no sense if there would have been cooling in such experiments. I fixed the error bars and now obtain a small interquartile range due to the reasons mentioned above. There are no more ensemble members with “negative” warming.

M3: It appears to me that the authors are trying hard to shape this work into newsworthy soundbites, but that it may be better suited in a more specialized journal where the results can be described in greater detail and the findings not oversold. An example of this overselling is the use of the “tipping element” framing, as discussed above. Another is the concluding paragraph of the manuscript which states:

“This implies that climate change and global warming would be self-amplifying in such scenarios and leaves the question if the direct interaction between tipping elements combined with the mean-field effect on GMT exerts positive feedbacks that are strong enough to trigger cascading transitions. This would, in turn, increase the risk of the Earth system moving towards a hothouse state.”

This statement is unjustified by the results of the work. There is no evidence presented that the different cryosphere components would strongly interact, as implied, as the study only considers the complete removal of each component or all at once and does not provide any assessment of the temperature at which a critical threshold would occur or, indeed, if any actually exist. It seems unlikely to me that the loss of one cryosphere component would be the sole trigger for the loss of another. And the concluding thought regarding moving the system towards a hothouse state is entirely unjustified. The results show, if anything, relatively small additional small additional warming from the loss of substantial cryosphere components that do not move the system to anywhere near a runaway greenhouse effect or a hothouse climate.

I urge the authors to read through the entire text to remove statements like this that are not supported by the results and to stick to reporting what has been done without the spin.

We agree with the referee that and removed statements like this from our conclusion to avoid overselling of our results. We also removed the Paris Agreement framing completely and toned down the tipping points framing by a more modest description of the changes in the cryosphere and why it is worth and important to investigate their temperature feedback on global mean temperature.

M4: It is implied in many places that the results have implications for the Paris targets and for warming over the 21st century. However, the results strictly apply only at equilibrium, which takes multiple millennia to be reached. This should be clarified in the abstract and elsewhere.

If implications for the Paris agreement are to be discussed, it should also be noted (on L112-114 and elsewhere) that the freshwater input from the loss of cryosphere components has the potential to completely cancel the warming you estimate on century timescales. For instance, recent papers estimate a global cooling of about 0.4K by 2100 from the freshwater input associated with loss of West Antarctic Ice Sheet (doi: 10.1038/s41586-019-0889-9, 10.1038/s41586-018-0712-z), which would more than offset the warming found here.

We reframed the part about the relevance for the Paris agreement since we agree with you that there are two dimensions that make our reference to the Paris agreement misinterpretable.

The first one is that the temperature feedbacks for the Arctic summer sea ice are already included in CMIP-5 simulations and match better with observations in the new generation of CMIP-6 simulations.

The second dimension are the other elements (Mountain glaciers, West Antarctic Ice Sheet, Greenland Ice Sheet) investigated in our study, where the time horizon of disintegration is clearly beyond 2100 with only smaller contributions beforehand.

Therefore we removed statements about the Paris agreement from our manuscript. Thank you for pointing us there (II 36-38 and 71-73).

Additionally we discuss the neglectance of freshwater input into the thermohaline circulation on decadal to centennial timescales more explicitly now quoting the studies you listed above. We agree that this is an important caveat that should be mentioned clearly, but does not impact the validity of our results on longer time scales (II 101-105).

Minor comments:

- L5-9: Right away you are implying that the cryosphere components you consider are those with critical thresholds that could be crossed, but that is incorrect (to our knowledge) in the case of summer sea ice and not supported here for the other components. Please reframe as discussed above.

In accordance to your suggestions in major point M1, we reframed this part of our manuscript. (II 4-10)

- L16 and L23: What are these numbers? Median or mean? Standard deviation or full ensemble spread?

These numbers are the median and the interquartile ranges in brackets now. Where applicable, we plotted box whiskers plots. This is also noted in the manuscript more clearly now (see II 13-14, II 115-116, Figs. 3, 4, S2, S3).

- L23 and elsewhere: Clarify that these values are specific to a given CO2 level and background warming. If CO2 is quadrupled and all Arctic summer sea ice is lost, then the loss of summer sea ice would cause no additional warming, right?

This is correct. We agree that this is important to mention this explicitly at this point in the manuscript (I 15).

- L56: I don't understand the use of the word "integrate" in this context.

This comment is resolved due to the recommended removal of passages with the Paris agreement.

- L79 and L147-148: The Antarctic Ice Sheet is an example of something that may truly have a tipping point but would not degrade quickly, thus not satisfying the "tipping element" definition used here.

Following your suggestion M1, we removed the framing on tipping elements.

- L147-148: How does this have any relevance for the Paris agreement, which aims to hold warming to less than 1.5K or 2K on decadal to centennial timescales? On timescales of millennia, other effects such as the long-term drawdown of CO2 by the oceans would become important, so it's not clear how these results would be relevant for temperature targets out that far.

We also found that our framing on the Paris agreement could be misleading such that we decided to tone this part down (compare to our response on major point M4).

- L157-159: This is the first place that you explain what the values in brackets represent. You also still have not said what the main estimate represents. Is it a median or mean, for instance?

The main value represents the median, the brackets the interquartile range and where applicable, we show box whiskers plots in the manuscript and supplement. This is now clearly stated where applicable to avoid misunderstanding (see II 13-14, II 115-116, Figs. 3, 4, S2, S3).

- L187-190 and Table 1: I do not understand why you have chosen to report values in W/m² rather than W/m²/K as before. The issue was not that reporting the values as a feedback was confusing or misleading, but instead that your values made no sense. Your response to

reviewers did not explain what had gone wrong so I am unable to assess whether the issue has been corrected or swept under the rug by a change in units. I think framing this in terms of feedbacks ($W/m^2/K$) as done previously is much more natural since the additional W/m^2 from the loss of the cryosphere components is difficult to compare with the other feedbacks acting.

We think that our use of the term “feedback” was misleading since we are computing two different settings that were both called feedbacks.

First: the usual notion of feedbacks is used in our calibration procedure (Fig. S1) that is applied in the 2xCO₂ experiments which were used as a filter method to select appropriate ensemble members. With the 39 selected ensemble members all further simulations are performed that are reported in the main manuscript and in the supplement.

Second: the values that are reported in Tab. 1 are a different kind of experiment. Here, we actually compute the additional radiative forcings at the top of the atmosphere that originate from the disappearance of the cryosphere elements at the same atmospheric CO₂ concentration. In Tab. 1 we report the values for the difference between disintegrated cryosphere elements and intact cryosphere, both at 400 ppm.

Similar investigations have been performed for the removal of Arctic sea ice. For a removal of one month during summer an additional radiative forcing of 0.3 W/m^2 is reported (Hudson, 2011, Journal of Geophysical Research: Atmospheres) which is in good agreement with Flanner et al. (2011, Nature Geoscience). We find a higher additional radiative forcing of $0.41+0.08 W/m^2 = 0.49 W/m^2$ (Tab. 1: Albedo + Clouds). This value is higher since we have low sea ice for approximately five months in our perturbed experiments instead of one as in Hudson, 2011 (II. 208-214).

So, we understand the confusion about the proper usage of the term “feedback” here. Now we agree that what is reported in Tab. 1 should not be called a feedback factor, but instead “additional radiative forcing”. Please find our major changes in II. 175-224.

However, in line with the referenced literature, we would like to keep the unit W/m^2 in our manuscript in order to avoid the earlier misunderstanding. However, we compiled the same table in $W/m^2/K$ below and hope that this helps to assess the correctness of our values. The values of the original table Tab. 1 in the manuscript have been used and divided by the warming due to disintegration of the respective element(s) to obtain the table Tab. 2 below. The quotients are ASSI = 0.19K, GIS = 0.13K, WAIS = 0.05K, MG = 0.08K and All = 0.43K.

Cryosphere element	LR + WV [W/m ² /K]	Clouds [W/m ² /K]	Albedo [W/m ² /K]	All changes [W/m ² /K]
ASSI	1.1 (0.9–1.2)	0.42 (0.37–0.47)	2.2 (1.8–2.5)	3.2 (3.1–4.2)
GIS	1.1 (1.0–1.2)	0.46 (0.38–0.54)	1.7 (1.5–1.9)	3.3 (3.0–3.6)
WAIS	1.0 (0.8–1.1)	0.8 (0.6–1.0)	2.0 (1.6–2.2)	3.6 (3.2–4.2)
MG	1.1 (1.0–1.3)	0.5 (0.4–0.6)	2.0 (1.8–2.1)	3.5 (3.3–4.0)
All	1.0 (0.9–1.1)	0.40 (0.37–0.44)	1.7 (1.5–1.8)	3.1 (2.8–3.4)

Table 2 | Drivers of additional warming as seen from the top of the atmosphere in [W/m²/K].

Please also compare this answer to a comment of reviewer #2 we had a similar question on the computation of feedbacks.

- L214: “exemplary” is the wrong word and could be removed.

We removed the word “exemplary”.

- L231-232: This does not follow from the results. This effect needs to be taken into account only if these cryosphere components are actually lost at the temperature values explored here. Yet no evidence is given that they will be.

In parallel to your major criticism M3, we removed this sentence to avoid unjustified overstating of our results (compare to our response on M3).

REVIEWER COMMENTS

Reviewer #1 (Remarks to the Author):

Dear authors,

Thanks for taking my comments into consideration. I think the paper is improving, also thanks to contributions of the two other reviewers.

I have a few things left.

Best wishes,

Martin Vancoppenolle

Two comments

1- Your title: Additional global warming due to loss of large ice masses.

This is problematic. Half of the temperature signal is due to sea ice, which is not a large ice mass.

Global warming is not "additional" (it is already in the current estimates). With such a title, people could believe you found a new, extra contribution to global warming.

2- I think the way you justify that you neglect freshwater feedbacks is contradictory and too complicated.

You say you focus on the long-term equilibrium temperature response, I'm not sure this is true.

The justification sentence you added to explain why you ignore freshwater feedbacks means everything and its opposite, I needed 5 minutes to understand it.

"Since we are interested in the long-term equilibrium temperature response of the Earth system, we neglect freshwater fluxes to the ocean which would occur when the cryosphere elements lose mass though in principle the new equilibrium state could depend on the model response to freshwater forcing, as for instance, on decadal to centennial time-scales for the West Antarctic and Greenland Ice Sheet"

I would suggest

(i) to reword into "For simplicity (or because our model does not allow to do so) we focus on purely radiative effects and neglect freshwater contribution to feedbacks, which has implications... (and detail why)"

(ii) Add in your abstract that your response estimates are in the high-range because freshwater feedbacks are ignored.

Best wishes

Reviewer #2 (Remarks to the Author):

[same as comments to editor] My remaining open questions have been comprehensively answered

in the rebuttal and have been clarified in the new version of the manuscript. In my original report I was in doubt which of two different interpretations of the phrase "change of feedback factors" had been applied for Tab.1. Now it is clear that the less "spectacular" interpretation is correct.

While the results are not very surprising in this now established interpretation, I still find it useful to give an systematic estimate how much warming a disappearance of the cryosphere elements causes at equilibrium. This type of study using systematic long-time simulations of carefully calibrated medium-complexity models may be one degree removed from the Paris policy debate about the next 100 years, but in my view it deserves a platform in Nature Communications.

Reviewer #3 (Remarks to the Author):

Third review of "Additional global warming due to loss of large ice masses" submitted to Nature Communications by Nico Wunderling, Matteo Willeit, Jonathan F. Donges, and Ricarda Winkelmann.

I appreciate the efforts the authors have made to revise the manuscript. It is much improved relative to the original submission. The manuscript is well written, and I believe I can now follow what was done at a level to be able to provide an assessment. I have several remaining concerns with the content of the manuscript which, if addressed, would make this suitable for publication in Nature Communications.

- I am still unable to understand how your radiative forcing values in Table 2 make sense in the context of your additional warming values. You report an additional radiative forcing of 1.35 W/m^2 for the loss of all cryosphere elements combined. For an equilibrium climate sensitivity of 3 K , the net climate feedback would be about $-1.33 \text{ W/m}^2/\text{K}$ assuming a forcing from CO_2 doubling of 4 W/m^2 . This would imply an additional warming of about $1.35/1.33 = 1 \text{ K}$. However, in the text you report a median additional warming of only about 0.4 K . What explains this discrepancy?

Perhaps the calculation should not include the ice albedo feedback since you are prescribing this as a forcing, in which case the additional warming would be about $1.35/(1.33+0.4) = 0.8 \text{ K}$. So that doesn't resolve the discrepancy. Perhaps the model has a different sensitivity to surface albedo changes than it does to CO_2 forcing. However, this doesn't seem to go in the right direction as all studies I know of suggest a higher sensitivity to forcing when it is localized at high latitudes (Stuecker et al. and many others).

This factor of two discrepancy seems important to resolve. Are your values in Table 2 correct? If so, why does your model show a much lower sensitivity to cryosphere forcing than to CO_2 forcing?

- The key result of this work hinges on the accurate representation of the additional radiative forcing arising from sea ice, glacier, or ice sheet loss. Once that additional forcing is established, the temperature response should follow from the fact that the model equilibrium climate sensitivity has been tuned to lie within the AR5 range while having feedbacks that lie within values found in comprehensive GCMs. The obvious question is then, does CLIMBER-2, with its idealized atmosphere, produce a realistic radiative response to these cryosphere changes? This would seem to require an accurate shortwave cloud climatology, an accurate representation of sea ice climatology, and possibly an accurate representation of the response of clouds to the ice loss. Petoukhov et al. (2000) shows that that version of CLIMBER-2 has a substantial planetary albedo biases at high latitudes, indicating potential biases in the response to ice loss.

Please provide some evidence that the version of CLIMBER-2 you are using is up to the task. For instance, how does its climatological planetary albedo (and partitioning between surface and atmosphere) compare against modern observations such as CERES? Or, how does the radiative forcing produced by ice loss compare against that from comprehensive GCMs, for instance as estimated from the many publicly available radiative kernels?

- The manuscript states the target range for the equilibrium climate was 1.5 to 4.5 K, consistent with the IPCC AR5 "likely" (68% probability) range. In the response to reviewers and Figure S1 it is shown that the 39 ensemble members selected span 2 to 3.75 K, which is somewhat narrower than the target range. This discrepancy should be noted in the text somewhere, perhaps in the methods section. Also, the narrow range of climate sensitivity used (2 to 3.75 K) should be noted as a reason for the narrow range of temperature responses to ice loss. If a broader climate sensitivity range was used, for instance 1 to 6 K to reflect the "very likely" (90%) range from AR5, then your warming uncertainties would be substantially larger.

Minor comments:

- L75: "on the other, the" is missing a word.

- L84 and elsewhere: To my knowledge there was no CMIP4, and the reference you refer to points to CMIP2. More generally, why use this reference from 15 years ago, when there are many new papers to choose from which provide updated estimates of radiative feedback ranges?

- L187: Why not 7%/degree from Clausius-Clapeyron scaling?

- L191: "clouds fast climate feedback" should read "cloud feedback".

- L191-194: Is this a description of the cloud changes that occur within CLIMBER-2? In nature and in more realistic models, there are a number of other cloud changes that can occur but which are not listed here (e.g., changes in cloud opacity or cloud altitude). Please clarify what cloud changes are possible in your model.

- L196: "clouds feedback" should read "cloud feedbacks".

- L211-214: You state that the reason you find a higher value is that you include several months of sea ice loss instead of one month as in Hudson (2011). This could be the right explanation, but is it obviously the right one? Hudson imposed sea ice loss on top of the present day climatology, while yours is on top of a 1.5 K world with less sea ice, so I might have expected a smaller forcing from your simulations. There is also the issue that your idealized model may not accurately represent the radiative response to ice loss, e.g., cloud climatology or cloud changes (see above).

- L229: What do you mean "not caused by the emissions of CO₂"? What else would cause the loss of the sea ice? I think you mean not caused by CO₂ within your model setup. You could just remove this phrase in the sentence to avoid confusion.

- L296: "an constraint" should read "a constraint".

- L297: Why accept such a large range of climatological sea ice extent compared to the observations?

For the authors' consideration:

I think I see what has gone wrong in your calculation of the radiative feedbacks given in Table 2 of the response to reviewers (units of $W/m^2/K$). The issue does not arise in the text since values there are reported in W/m^2 . But I'll illustrate the issue here in case it is useful for you.

Table 2 shows changes in individual feedbacks when cryosphere elements are lost. The values are far too large. One way to see this is that for an equilibrium climate sensitivity of 3 K, the net climate feedback (sum of Planck, LR+WV, cloud, albedo) would be about $-1.33 W/m^2/K$. A feedback change of $3 W/m^2/K$ as reported in the Table would produce a net feedback that is greater than zero, i.e., formally a runaway climate, which is not seen in your simulations. Where have you gone wrong in your calculations?

Consider two feedbacks, λ_{1} (baseline simulation) and λ_{2} (cryosphere loss simulation), with their difference $\lambda' = \lambda_{2} - \lambda_{1}$. I believe you are trying to report λ' in Table 2. You could calculate it by directly calculating $\lambda_{1} = R/T$ and $\lambda_{2} = (R + R')/(T + T')$ and then take their difference, where R and T are the baseline global radiative and temperature response and R' and T' are the changes in radiation and temperature induced by cryosphere loss (from Table 1).

This would be equivalent to: $\lambda' = (R + R')/(T + T') - R/T = (R' - R \cdot T'/T) / (T + T')$. For small T' , this is approximately $\lambda' = R'/T$.

However, I believe the calculation you've done for Table 2 is $\lambda' = R'/T'$, which would be incorrect but would explain the unphysically-large λ' values.

Dear Mrs Plail, Dear Reviewers,

We are grateful that our manuscript is of interest to Nature Communications in the view of the reviewers. We appreciate the current reviews that help us to clarify and improve our manuscript.

The main changes are:

1. Title: We changed the title to avoid confusion about the cryosphere elements that we are investigating.
2. Feedbacks: We worked on the explanation of the values in Tab. 1. These values can be interpreted as feedback values for water vapour, lapse rate and cloud feedbacks with very small forcing contributions, but for albedo feedbacks they are a feedback and a forcing (due to the removal of cryosphere elements in our experiments).
3. Clouds: We compared the planetary albedo representation of clouds to CERES data and found that shortwave cloud radiative effects are well represented by CLIMBER-2. Here, we performed additional simulations and added a supplementary figure to the manuscript.

We are thankful for the opportunity to further improve our manuscript and have carefully included the comments of the reviewers into the manuscript. The point-by-point response can be found below and the changes in the manuscript are colour marked in blue.

Sincerely yours,

Nico Wunderling, Matteo Willeit, Jonathan F. Donges & Ricarda Winkelmann

Reviewer #1 (Remarks to the Author):

Dear authors,

Thanks for taking my comments into consideration. I think the paper is improving, also thanks to contributions of the two other reviewers.

I have a few things left.

Best wishes,

Martin Vancoppenolle

Two comments

1- Your title: Additional global warming due to loss of large ice masses.

This is problematic. Half of the temperature signal is due to sea ice, which is not a large ice mass.

Global warming is not "additional" (it is already in the current estimates). With such a title, people could believe you found a new, extra contribution to global warming.

We agree with the reviewer that this title could cause confusion. Thus, we changed it to “Global warming commitment due to loss of large ice masses and Arctic summer sea ice” omitting the word additional and explicitly mentioning the Arctic summer sea ice.

2- I think the way you justify that you neglect freshwater feedbacks is contradictory and too complicated.

You say you focus on the long-term equilibrium temperature response, I'm not sure this is true.

The justification sentence you added to explain why you ignore freshwater feedbacks means everything and its opposite, I needed 5 minutes to understand it.

"Since we are interested in the long-term equilibrium temperature response of the Earth system, we neglect freshwater fluxes to the ocean which would occur when the cryosphere elements lose mass though in principle the new equilibrium state could depend on the model response to freshwater forcing, as for instance, on decadal to centennial time-scales for the West Antarctic and Greenland Ice Sheet"

I would suggest

(i) to reword into "For simplicity (or because our model does not allow to do so) we focus on purely radiative effects and neglect freshwater contribution to feedbacks, which has implications... (and detail why)"

(ii) Add in your abstract that your response estimates are in the high-range because freshwater feedbacks are ignored.

We agree that this sentence is difficult to understand and we have rewritten this paragraph as you suggested also mentioning the transient influence of freshwater input (see II 100-104 and I 13).

Furthermore, it is true that freshwater input would offset/reduce the warming contribution as long as it is flowing into the AMOC, but this does not hold anymore when the ice sheets are disintegrated and further freshwater input comes to a halt. In our manuscript, we are looking at equilibrium responses (after 10,000 simulation years) when the large ice masses are assumed to be disintegrated. Thus, we do not think that our warming estimates would be in the high range, unless the AMOC irreversibly collapses. However, in this case the AMOC goes approximately back to its original strength when freshwater input stops (Rahmstorf et al., 2005, GRL, *Thermohaline circulation hysteresis: A model intercomparison*).

Best wishes

Reviewer #2 (Remarks to the Author):

[same as comments to editor] My remaining open questions have been comprehensively answered in the rebuttal and have been clarified in the new version of the manuscript. In my original report I was in doubt which of two different interpretations of the phrase "change of feedback factors" had been applied for Tab.1. Now it is clear that the less "spectacular" interpretation is correct.

While the results are not very surprising in this now established interpretation, I still find it useful to give a systematic estimate how much warming a disappearance of the cryosphere elements causes at equilibrium. This type of study using systematic long-time simulations of carefully calibrated medium-complexity models may be one degree removed from the Paris policy debate about the next 100 years, but in my view it deserves a platform in Nature Communications.

Reviewer #3 (Remarks to the Author):

Third review of “Additional global warming due to loss of large ice masses” submitted to Nature Communications by Nico Wunderling, Matteo Willeit, Jonathan F. Donges, and Ricarda Winkelmann.

I appreciate the efforts the authors have made to revise the manuscript. It is much improved relative to the original submission. The manuscript is well written, and I believe I can now follow what was done at a level to be able to provide an assessment. I have several remaining concerns with the content of the manuscript which, if addressed, would make this suitable for publication in Nature Communications.

We thank the referee for this positive assessment.

- I am still unable to understand how your radiative forcing values in Table 2 make sense in the context of your additional warming values. You report an additional radiative forcing of 1.35 W/m^2 for the loss of all cryosphere elements combined. For an equilibrium climate sensitivity of 3 K, the net climate feedback would be about $-1.33 \text{ W/m}^2/\text{K}$ assuming a forcing from CO₂ doubling of 4 W/m^2 . This would imply an additional warming of about $1.35/1.33 = 1 \text{ K}$. However, in the text you report a median additional warming of only about 0.4 K. What explains this discrepancy?

Perhaps the calculation should not include the ice albedo feedback since you are prescribing this as a forcing, in which case the additional warming would be about $1.35/(1.33+0.4) = 0.8 \text{ K}$. So that doesn't resolve the discrepancy. Perhaps the model has a different sensitivity to surface albedo changes than it does to CO₂ forcing. However, this doesn't seem to go in the right direction as all studies I know of suggest a higher sensitivity to forcing when it is localized at high latitudes (Stuecker et al. and many others).

This factor of two discrepancy seems important to resolve. Are your values in Table 2 correct? If so, why does your model show a much lower sensitivity to cryosphere forcing than to CO₂ forcing?

Thank you for your detailed comments here and also for the following rough calculations and considerations on the values in Table 1. You can find the details of our answer after the following section below.

For the authors' consideration:

I think I see what has gone wrong in your calculation of the radiative feedbacks given in Table 2 of the response to reviewers (units of $\text{W/m}^2/\text{K}$). The issue does not arise in the text since values there are reported in W/m^2 . But I'll illustrate the issue here in case it is useful for you.

Table 2 shows changes in individual feedbacks when cryosphere elements are lost. The values are far too large. One way to see this is that for an equilibrium climate sensitivity of 3 K, the net climate feedback (sum of Planck, LR+WV, cloud, albedo) would be about $-1.33 \text{ W/m}^2/\text{K}$. A feedback change of $3 \text{ W/m}^2/\text{K}$ as reported in the Table would produce a net feedback that is greater than zero, i.e., formally a runaway climate, which is not seen in your simulations. Where have you gone wrong in your calculations?

Consider two feedbacks, λ_{1} (baseline simulation) and λ_{2} (cryosphere loss simulation), with their difference $\lambda' = \lambda_{2} - \lambda_{1}$. I believe you are trying to report λ' in Table 2. You could calculate it by directly calculating $\lambda_{1} = R/T$ and $\lambda_{2} = (R + R')/(T + T')$ and then take their difference, where R and T are the baseline global radiative and temperature response and R' and T' are the changes in radiation and temperature induced by cryosphere loss (from Table 1).

This would be equivalent to: $\lambda' = (R + R')/(T + T') - R/T = (R' - R \cdot T'/T) / (T + T')$. For small T' , this is approximately $\lambda' = R'/T$.

However, I believe the calculation you've done for Table 2 is $\lambda' = R'/T'$, which would be incorrect but would explain the unphysically-large λ' values.

The values reported in Table 1 represent the change in net radiation at the top of the atmosphere resulting from prescribing the respective climate fields from the perturbed experiments in the control simulation and are computed from offline double calls to the radiation subroutines in the model. These values include both the forcing AND the feedbacks, e.g. for surface albedo: $F_{\text{alb}} + \lambda_{\text{alb}} \cdot \Delta T_{\text{eq}}$, where F_{alb} is the direct radiative forcing from albedo changes originating from the removal of the cryosphere components and ΔT_{eq} is the equilibrium global temperature change. The reason why we are reporting the values this way is that in our experiments surface albedo acts both as a forcing (we are prescribing changes in the surface type) and as a feedback (surface albedo will also change as a response to e.g. additional warming).

Assuming that λ in our experiments is not very different from the λ for CO2 doubling, the equation $F + \lambda \cdot \Delta T_{\text{eq}} = 0$, where F is the direct radiative forcing originating from the removal of the cryosphere components, can actually be used to get a rough estimate of the forcing. For $\Delta T_{\text{eq}} = 0.43 \text{ K}$ and $\lambda = -1.33 \text{ W/m}^2/\text{K}$ this gives $F = 0.57 \text{ W/m}^2$. If this forcing estimate is removed from the value for albedo reported in Table 1 (0.72 W/m^2), an estimate for the albedo feedback can be derived: $(0.72 - 0.57)/0.43 \text{ W/m}^2/\text{K} = 0.35 \text{ W/m}^2/\text{K}$, which is fully consistent with the albedo feedback parameter reported for CLIMBER-2 and other GCMS for CO2 doubling in Fig. S1a.

On the other hand, the values for water vapour + lapse rate and clouds in Table 1 can actually to a very good approximation be interpreted directly as feedback parameters once they are divided by ΔT_{eq} .

In the revised version of the manuscript, we included a short discussion of these points to make it more transparent and avoid misinterpretations of the presented values. Changes can be found in II 178-185. Furthermore, we replaced “radiative forcing” by “radiative perturbation (at the top of the atmosphere)” to avoid misinterpretation throughout the manuscript.

- The key result of this work hinges on the accurate representation of the additional radiative forcing arising from sea ice, glacier, or ice sheet loss. Once that additional forcing is established, the temperature response should follow from the fact that the model equilibrium climate sensitivity has been tuned to lie within the AR5 range while having feedbacks that lie within values found in comprehensive GCMs. The obvious question is then, does CLIMBER-2, with its idealized atmosphere, produce a realistic radiative response to these cryosphere changes? This would seem to require an accurate shortwave cloud climatology, an accurate representation of sea ice climatology, and possibly an accurate representation of the response of clouds to the ice loss. Petoukhov et al. (2000) shows that that version of CLIMBER-2 has a substantial planetary albedo biases at high latitudes, indicating potential biases in the response to ice loss.

Please provide some evidence that the version of CLIMBER-2 you are using is up to the task. For instance, how does its climatological planetary albedo (and partitioning between surface and atmosphere) compare against modern observations such as CERES? Or, how does the radiative forcing produced by ice loss compare against that from comprehensive GCMs, for instance as estimated from the many publicly available radiative kernels?

We agree that it is important to mention why CLIMBER-2 is an appropriate model for our study. Generally, the use of an Earth system model of intermediate complexity is a good choice since it is computationally efficient and can thus be used to compute ensemble runs with a length of several thousand years which is more difficult with global circulation models due to computational constraints (see II 256-260).

Furthermore, we compared the planetary albedo from our CLIMBER-2 control ensemble with current CERES observations during the JJA and the DJF months (see Figure 1). We find that the general pattern of planetary albedo is quite similar between CERES and CLIMBER-2, albeit minor differences for example in very high latitudes exist. The zonal mean of the CLIMBER-2 planetary albedo values also shows very small deviations from the CERES data (see Figure 2). We added a discussion on that in the manuscript, see II 251-256 and added Figure 2 as supplementary Figure S5.

In Figure 3, we also compare the CLIMBER-2 surface albedo shortwave radiative kernels at the surface and at the top of the atmosphere (TOA) with the available radiative kernels of three general circulation models. The figure shows that the CLIMBER-2 albedo kernels are generally in good agreement with the kernels of more complex GCMs.

Figure 1: Planetary albedo during DJF and JJA. The left column represents the CLIMBER-2 simulations and the right column the CERES observations.

Figure 2: Planetary albedo during June, July and August (panel a) and during December, January and February (panel b). The CLIMBER-2 data is shown as the full ensemble spread, where the straight line indicates the median of the ensemble, the dashed line the interquartile range and the shaded area the full ensemble spread. The data is cut at 65°S during summer since no planetary albedo values are available at this time due to the Antarctic night. The CERES data deviates at this edge from the CLIMBER data due to the observational sparse data close to the Antarctic night region. For the same reason, the data is cut due to the Arctic night in panel b. This figure is added to the supplement of the paper as Fig. S5.

Figure 3: Radiative kernels for the surface (top panel) and surface (bottom panel), where CLIMBER-2 is compared to available kernels from GCM simulations.

- The manuscript states the target range for the equilibrium climate was 1.5 to 4.5 K, consistent with the IPCC AR5 “likely” (68% probability) range. In the response to reviewers and Figure S1 it is shown that the 39 ensemble members selected span 2 to 3.75 K, which is somewhat narrower than the target range. This discrepancy should be noted in the text somewhere, perhaps in the methods section. Also, the narrow range of climate sensitivity used (2 to 3.75 K) should be noted as a reason for the narrow range of temperature responses to ice loss. If a broader climate sensitivity range was used, for instance 1 to 6 K to reflect the “very likely” (90%) range from AR5, then your warming uncertainties would be substantially larger.

With all our constraints that we applied, potential ensemble members with higher (or lower) equilibrium climate sensitivity have been filtered out. We noted in the manuscript that what you mention above might be a reason for a narrower climate response than would be expected if the equilibrium climate response would cover all values from 1.5 to 4.5 K (see II 314-316).

Minor comments:

- L75: “on the other, the” is missing a word.

Thanks.

- L84 and elsewhere: To my knowledge there was no CMIP4, and the reference you refer to points to CMIP2. More generally, why use this reference from 15 years ago, when there are many new papers to choose from which provide updated estimates of radiative feedback ranges?

We used the values that have been reported in AR4 from Soden & Held (2006; not as erroneously stated from CMIP-4, we corrected that in the manuscript) instead of AR5 since the values reported in Soden & Held are still valid since they agree well with the estimates from the new models used in AR5, except for cloud feedbacks. But cloud feedbacks are not very well constrained in general, so the “true” cloud feedback strength might even lay outside the given interval in AR5 or Soden & Held.

Thus, apart from a less well constrained cloud feedback in the CMIP-5 ensemble, the changes in the other feedback types (Water vapour + Lapse rate, Albedo) are very small from AR4 to AR5. This can be compared in Figure 9.43 in the IPCC report (on page 819), where the values from CMIP-5 and Soden & Held (2006) are directly plotted.

We included this in the methods of our manuscript in II 290-294.

- L187: Why not 7%/degree from Clausius-Clapeyron scaling?

We have used 5% since this is the value that has been reported in Gregory & Huybrechts (2006, see: <https://doi.org/10.1098/rsta.2006.1796>) for climate change simulations on Greenland and Antarctica. But, we agree that we should refer to the Clausius-Clapeyron equation. We replaced the respective phrase as you suggested. (see II 194-196)

- L191: “clouds fast climate feedback” should read “cloud feedback”.

Thank you, we replaced clouds by cloud feedback.

- L191-194: Is this a description of the cloud changes that occur within CLIMBER-2? In nature and in more realistic models, there are a number of other cloud changes that can occur but which are not listed here (e.g., changes in cloud opacity or cloud altitude). Please clarify what cloud changes are possible in your model.

Apart from changes in cloud fraction, cloud top height can change following changes in the height of the tropopause. The cloud optical thickness parameterisation includes a dependence on the cumulus cloud fraction in addition to a prescribed increase of optical thickness with latitude (see II 251-256). We removed lines 191-194 to avoid confusion.

- L196: “clouds feedback” should read “cloud feedbacks”.

Thank you, this is replaced.

- L211-214: You state that the reason you find a higher value is that you include several months of sea ice loss instead of one month as in Hudson (2011). This could be the right explanation, but it is it obviously the right one? Hudson imposed sea ice loss on top of the present day climatology, while yours is on top of a 1.5 K world with less sea ice, so I might have expected a smaller forcing from your simulations. There is also the issue that your idealized model may not accurately represent the radiative response to ice loss, e.g., cloud climatology or cloud changes (see above).

We looked into the cloud representation in CLIMBER-2 and hope we could resolve potential issues with a misrepresentation of clouds (see above). We agree that this sentence should not be formulated as a definite statement due to possible model uncertainties as well as minor differences in the background CO2 concentration (Hudson used the sea ice cover from 2007 at around 1.0°C above pre-industrial, whereas we used 1.5°C above pre-industrial). We reframed this sentence in II 212-215.

- L229: What do you mean “not caused by the emissions of CO₂”? What else would cause the loss of the sea ice? I think you mean not caused by CO₂ within your model setup. You could just remove this phrase in the sentence to avoid confusion.

We agree and removed this sentence.

- L296: “an constraint” should read “a constraint”.

Thank you, this is corrected.

- L297: Why accept such a large range of climatological sea ice extent compared to the observations?

The reason for our constraint is that we would like to cover a large range of different simulations within our ensemble that are also able to cover the uncertainties in the other feedback parameters such as cloud, albedo, water vapour, lapse rate, equilibrium climate sensitivity, etc. (see Fig. S1). On the other hand, our ensemble shows Arctic summer sea ice areas of around 3 to 4.5x10⁶ km² such that we can be sure that outlier values are not overemphasized (see interquartile range in September, blue box in supplementary Fig. S3), only the full range of ensemble members is a bit larger. A smaller spread of sea ice would also reduce the spread of all our results in general as well. Then, our results would appear more accurate than would be realistic.

Furthermore, the spread of the minimal northern hemisphere sea ice cover is on the order of 4x10⁶ km² and as such in the same range as our spread of 5x10⁶km². However, the minimal CLIMBER-2 sea ice area is lower than for the CMIP5 global circulation models under pre-industrial conditions (see Howell et al. (2016), doi:10.5194/cp-12-749-2016 for CMIP5 sea ice). Thus, the warming effect through sea ice removal is on the conservative side in this study.

REVIEWERS' COMMENTS:

Reviewer #3 (Remarks to the Author):

The authors have addressed my concerns and I recommend publication.

Below are several additional suggestions for the authors, including some that may help avoid some areas of potential confusion for readers.

- L24-26: This sentence would more precisely be "Our findings have implications for the global climate since the disintegration of cryosphere elements can lead to a further rise of the global mean temperature on long time scales." I say this because the term 'temperature feedbacks' often refers to the Planck and lapse-rate feedbacks, so as written your sentence is somewhat confusing.

- L47: You may consider citing another recent paper on this topic here as well, doi: 10.1017/jog.2018.57

- L85-87: I still think it would be important to state the ECS range for your ensemble here (2-3.75K). As written, the text implies that you have a range of ECS matching some GCM ensemble. Readers would naturally think your ECS range is either 2-4.7K from matching CMIP5 or perhaps 1.8-5.8K from CMIP6, for instance. In fact, your ECS range is substantially more narrow.

- L121-122: Again, I'm finding myself confused because this language regarding feedbacks is imprecise. This additional warming of 29% is due to the albedo change associated with your prescribed disintegration of GIS and WAIS (a forcing), not "due to the ice-climate feedbacks" which would, in my mind, refer to an ice albedo feedback in response to those prescribed forcings. If it is the disintegration of GIS and WAIS itself that you are referring to here as "ice-climate feedbacks", then you should state this. Otherwise, you could drop "due to the ice-climate feedbacks" in this sentence.

- Figure 3 caption: You say you use a least-squares fit to convert CO2 concentration to temperature. What is this based on? Your simulations of temperature change at different CO2 levels without your prescribed cryosphere changes?

- L181-184 and Table 1: Thanks to your response to my previous comments, I think I finally understand how to interpret the relationship between your forcing values (W/m^2) and temperature changes. I think a key thing that was (and remains) confusing is the idea that there will be changes in surface albedo in responses to the temperature change induced by your imposed sea ice loss (a feedback), since without any more sea ice I was anticipating no more sea ice albedo feedback. I think the key to understanding this is that there is still a snow albedo feedback which responds to surface temperature changes. The role of the snow albedo feedback would be good to clarify in the text so readers can follow this subtle argument. It may also be worth stating in the Table 1 caption that the albedo 'forcing' values (W/m^2) refer to the combined effect of sea ice removal and the radiative effect of snow loss in response to the additional warming.

Dear Mrs Plail, Dear Reviewers,

We are very happy that our manuscript is recommended for publication at Nature Communications from all reviewers. We are very thankful for the comments from all three reviewers to improve our manuscript substantially throughout the whole review process. Below, we provide a point-by-point answer to the remaining issues that were raised by the reviewers. We also specify the parts in the manuscript where these changes can be found.

Best regards,

Nico Wunderling, Matteo Willeit, Jonathan F. Donges & Ricarda Winkelmann

Reviewer #3 (Remarks to the Author):

The authors have addressed my concerns and I recommend publication.

We thank the reviewer for this overall positive evaluation.

Below are several additional suggestions for the authors, including some that may help avoid some areas of potential confusion for readers.

- L24-26: This sentence would more precisely be “Our findings have implications for the global climate since the disintegration of cryosphere elements can lead to a further rise of the global mean temperature on long time scales.” I say this because the term ‘temperature feedbacks’ often refers to the Planck and lapse-rate feedbacks, so as written your sentence is somewhat confusing.

Thank you. We did not intend to refer to Planck or lapse-rate feedbacks here. We changed this sentence accordingly (page 2 last sentence).

- L47: You may consider citing another recent paper on this topic here as well, doi: 10.1017/jog.2018.57

We added this additional reference on the current disequilibrium of many mountain glaciers to our manuscript, see page 3 (reference 13).

- L85-87: I still think it would be important to state the ECS range for your ensemble here (2-3.75K). As written, the text implies that you have a range of ECS matching some GCM ensemble. Readers would naturally think your ECS range is either 2-4.7K from matching CMIP5

or perhaps 1.8-5.8K from CMIP6, for instance. In fact, your ECS range is substantially more narrow.

We agree with the reviewer and we now state that not only in the methods section as before, but also in the main manuscript (page 5 at the end of the 2nd paragraph).

- L121-122: Again, I'm finding myself confused because this language regarding feedbacks is imprecise. This additional warming of 29% is due to the albedo change associated with your prescribed disintegration of GIS and WAIS (a forcing), not "due to the ice-climate feedbacks" which would, in my mind, refer to an ice albedo feedback in response to those prescribed forcings. If it is the disintegration of GIS and WAIS itself that you are referring to here as "ice-climate feedbacks", then you should state this. Otherwise, you could drop "due to the ice-climate feedbacks" in this sentence.

We concord that our notion of "ice-climate feedbacks" in this line was confusing since the temperature response of 29% of additional warming refers to both, the forcing and the associated feedbacks. For clarity, we dropped the respective half-sentence "due to the ice-climate feedbacks".

- Figure 3 caption: You say you use a least-squares fit to convert CO₂ concentration to temperature. What is this based on? Your simulations of temperature change at different CO₂ levels without your prescribed cryosphere changes?

Yes, it is based on the respective temperature that originates from the global mean temperature in CLIMBER-2 at different levels of CO₂ concentration without prescribed changes in the cryosphere (using the full ensemble). Since the relationship between CO₂ concentration and increase of the global mean temperature (Δ GMT) is slightly nonlinear, we used python's least-squares fit function `scipy.optimize.curve_fit`. We clarified that in the manuscript in the caption of Figure 3.

- L181-184 and Table 1: Thanks to your response to my previous comments, I think I finally understand how to interpret the relationship between your forcing values (W/m^2) and temperature changes. I think a key thing that was (and remains) confusing is the idea that there will be changes in surface albedo in responses to the temperature change induced by your imposed sea ice loss (a feedback), since without any more sea ice I was anticipating no more sea ice albedo feedback. I think the key to understanding this is that there is still a snow albedo feedback which responds to surface temperature changes. The role of the snow albedo feedback would be good to clarify in the text so readers can follow this subtle argument. It may also be worth stating in the Table 1 caption that the albedo 'forcing' values (W/m^2) refer to the combined effect of sea ice removal and the radiative effect of snow loss in response to the additional warming

We agree and directly mention that surface changes are allowed after the artificial removal of the cryosphere components. This feedback component comprises changes in the vegetation or the snow cover in response to the warming from the forcing component (page 13 [lines 10-14] and caption of Table 1).